# Semi-Truths: A Large-Scale Dataset of AI-Augmented Images for Evaluating Robustness of AI-Generated Image detectors

**Anisha Pal**[1*]   **Julia Kruk**[1*]   **Mansi Phute**[1]   **Manognya Bhattaram**[1]
**Diyi Yang**[2]   **Duen Horng Chau**[1]   **Judy Hoffman**[1]

[1]Georgia Institute of Technology, [2]Stanford University
{apal72, jkruk3, mphute6, polo, judy}@gatech.edu
{manognya.work}@gmail.com , {diyiy}@stanford.edu
https://huggingface.co/datasets/semi-truths/Semi-Truths

**Visualizing Varying Magnitudes of Change by Area Ratio (AR)**

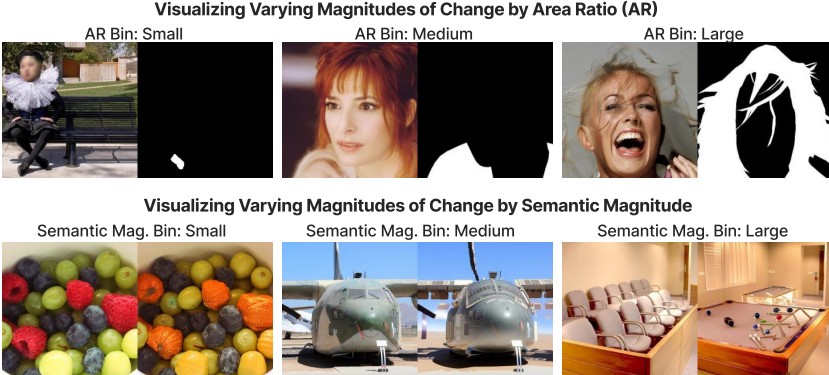

Figure 1: SEMI-TRUTHS image augmentations that are measured by the size of the augmented region (Area Ratio) and the semantic change achieved (Semantic Magnitude), categorized into 3 levels - small (col1), medium (col2), and large (col3).

## Abstract

Text-to-image diffusion models have impactful applications in art, design, and entertainment, yet these technologies also pose significant risks by enabling the creation and dissemination of misinformation. Although recent advancements have produced AI-generated image detectors that claim robustness against various augmentations, their true effectiveness remains uncertain. Do these detectors reliably identify images with different levels of augmentation? Are they biased toward specific scenes or data distributions? To investigate, we introduce SEMI-TRUTHS, featuring $27,600$ real images, $223,400$ masks, and $1,329,155$ AI-augmented images that feature targeted and localized perturbations produced using diverse augmentation techniques, diffusion models, and data distributions. Each augmented image is accompanied by metadata for standardized and targeted evaluation of detector robustness. Our findings suggest that state-of-the-art detectors exhibit varying sensitivities to the types and degrees of perturbations, data distributions, and augmentation methods used, offering new insights into their performance and limitations. The code for the augmentation and evaluation pipeline is available at https://github.com/J-Kruk/SemiTruths.

---

[*]Equal contribution.

38th Conference on Neural Information Processing Systems (NeurIPS 2024) Track on Datasets and Benchmarks.

# 1 Introduction

The rise of text-to-image generative models has democratized automated image creation for machine learning practitioners and the general public alike. While existing architectures like Variational Autoencoders [84, 32] and GANs [4, 100, 30, 35, 38] have produced realistic images for several years, diffusion models [15, 69, 13] have enhanced image quality, diversity, and usability, driving their rapid adoption. However, this technology presents a double-edged sword: despite its applications in fields like art, design, marketing, and entertainment [34, 96], its growing ubiquity brings a heightened risk of misuse for spreading misinformation [94, 55]. Recent incidents reveal an alarming rise in AI-modified images being used to perpetrate harmful acts such as misinformation campaigns [19, 91, 95], fraud, defamation, and identity theft [26, 79, 21, 9]. One concerning capability of these models is their ability to alter small attributes of an original image. Rather than creating images from scratch, individuals can alter only specific parts or attributes of an image to drastically change the narrative while decreasing the likelihood of detection. An example of one such "semi-truth" is the "Sleepy Joe" [72] video circulated on Twitter in 2020, where President Joe Biden's face was edited to appear as if he fell asleep during an interview. The implications of such subtle perturbations and their potential to spread misinformation [19, 91, 95] underscore the critical need for automated detection of such attacks.

| Dataset | Magnitude of Change | Targeted Perturb. | Saliency Check | Data Collection | Generation | | | Data Dist. | | Scale | |
|---|---|---|---|---|---|---|---|---|---|---|---|
| | | | | | GANs | Diffusion | #Methods | Scene | #Real Bench. | Real | Fake |
| 1 DFDC [16] | ✗ | ✗ | ✗ | Generated | ✓ | ✗ | 8 | Face | 1 | 488.4k | ∼1.7M |
| 2 FaceForensics++ [71] | ✗ | ✗ | ✗ | Generated | ✓ | ✗ | 4 | Face | 1 | 509.9k | ∼1.8M |
| 3 Celeb-DF [47] | ✗ | ✗ | ✓ | Generated | ✓ | ✗ | 1 | Face | 1 | 225.4k | ∼2.1M |
| 4 DeepFakeFace [76] | ✗ | ✗ | ✗ | Generated | ✗ | ✓ | 3 | Face | 1 | 30k | 90k |
| 5 CIFAKE [5] | ✗ | ✗ | ✗ | Generated | ✗ | ✓ | 1 | General | 1 | 60k | 60k |
| 6 DiffusionDB [90] | ✗ | ✗ | ✓ | Sourced | ✗ | ✓ | 1 | General | 0 | 0 | 14M |
| 7 MidJourney prompts [83] | ✗ | ✗ | ✗ | Sourced | ✗ | ✓ | 1 | General | 0 | 0 | 248k |
| 8 TWIGMA [10] | ✗ | ✗ | ✗ | Sourced | ✗ | ✓ | unknown | General | 0 | 0 | 800k |
| 9 GenImage [102] | ✗ | ✗ | ✗ | Generated | ✓ | ✓ | 8 | General | 1 | 1.33M | 1.35M |
| 10 SEMI-TRUTHS | ✓ | ✓ | ✓ | Generated | ✗ | ✓ | 8 | General | 6 | 27,635 | ∼1.33M |

Table 1: **SEMI-TRUTHS vs other AI-generated image datasets.** We compare SEMI-TRUTHS with other AI-generated image datasets across multiple categories: (1) Magnitude of Change: provides metadata on the magnitude of perturbations; (2) Targeted Perturb.: performs targeted perturbation of images; (3) Saliency Check: saliency assessment of fake images; (4) Data Collection: data collection strategy, *Generated* or *Sourced* from publicly available portals; (5) Generation: generator category and number of methods used (TWIGMA's method was unknown since its images were sourced from Twitter); (6) Data Distribution: scene variation and diversity of real benchmarks; (7) Scale: number of real and fake images.

However, existing datasets for training and evaluating AI-generated image detectors primarily consist of fully synthesized images, often limited to human faces [16, 71, 47, 39, 14]. This narrow focus fails to capture the diversity of real-world perturbations and does not reveal model biases toward different degrees of change. To address this, we introduce SEMI-TRUTHS, which includes AI-augmented images with varying levels of perturbation (detailed comparison in Tab. 1), enabling the evaluation of detectors against more realistic and diverse attacks like the "Sleepy Joe" video [72].

To develop a resource for stress-testing specific biases in AI-generated image detectors, precise control over the nature and extent of changes is essential. To achieve this, we source images from 6 popular semantic segmentation datasets, which contain salient images and labeled entity masks across a range of subjects. We categorize the magnitude of change in SEMI-TRUTHS based on two criteria: (1) the size of the augmented region, and (2) the semantic change achieved. The size of the augmented region is captured by surface area of the mask, structural similarity index measure (SSIM), mean squared error (MSE), and a custom metric derived from MSE (see Algo. 2). Semantic changes are quantified using metrics such as Semantic Magnitude, Learned Perceptual Image Patch Similarity (LPIPS) [98], DreamSim [22], and Sentence Similarity [78] (see Sec. 3.1). Perturbed images in SEMI-TRUTHS are generated through diffusion inpainting and prompt-based-editing [28, 54] techniques using 5 different diffusion algorithms [63, 74, 61, 70].

Our approach to curating SEMI-TRUTHS employs a flexible, plug-and-play framework for human-guidance-free image editing. This pipeline is designed to ensure reusability and adaptability to new data distributions, large language models for prompt perturbation, diffusion models, and various image synthesis techniques. By releasing this framework, we aim to empower the community to create customized stress tests to evaluate AI-generated image detectors for specific use cases.

Finally, we demonstrate how the knowledge abstractions in SEMI-TRUTHS can be used to identify the sensitivities of existing detectors. By stress-testing 6 models, we reveal unique sensitivities to different data distributions, diffusion models, and perturbation degrees. Our goal is to offer a resource for targeted, interpretable, and standardized evaluation of AI-generated image detection systems, and to provide a customizable evaluation pipeline for the community.

## 2 Related Work

**AI Augmented Image dataset**    The field of AI-based image generation and perturbation has rapidly evolved from autoencoders [20] and graphics-based techniques [81] to GANs [101, 58, 2, 49, 6] and, more recently, diffusion models [57, 70, 61, 24]. These advancements have heightened ethical concerns regarding identity theft and misinformation, [3, 27, 31] necessitating robust datasets for AI-generated image detection. While most research has focused on GAN-generated human faces [16, 71, 47, 39, 14], there is a growing emphasis on diffusion-based techniques for detection of deepfakes [76], digital forgery [75] and generic AI-generated content [102, 5, 83, 90]. However, existing datasets face several limitations that restrict their applicability as a benchmark for developing robust detection systems. They often come from a single model [83, 90] or source data distribution [102, 5], lack detailed generation and image metadata [10], and provide limited control over degree and quality of perturbations [83, 90, 102, 5, 76, 10, 66]. Furthermore, they do not offer scalable pipelines for integrating future image generation and perturbation techniques and are limited in their analysis of detection methods. Recognizing these gaps, we introduce SEMI-TRUTHS that incorporates multiple model variations, perturbing techniques, and source data distributions, provides comprehensive metadata, and offers fine-grained control over the quality and degree of perturbations (Tab. 1 summarizes SEMI-TRUTHS's contributions).

**Image editing pipelines**    With the advent of diffusion models, the field of image editing has seen tremendous advancements [33]. Recent developments in image inpainting, both in text-conditioned [92, 93, 87, 97] and unconditioned [51] settings, have enabled fine-grained control over image editing significantly enhancing precision and quality. While image inpainting requires the use of masks, prompt-based image editing [28, 54] performs targeted perturbations conditioned solely on text prompts. Existing frameworks like LANCE [62] and InstructPix2Pix [7] leverage this capability to develop automated image perturbation pipelines. LANCE [62], leveraging large language models (LLMs)[82] and image captioning[45], enables human-supervision-free image editing across diverse perturbations. Building on this, we extend LANCE [62] to handle a broader range of perturbation magnitudes, guided by semantic change definitions [8, 36]. Our approach integrates LlaVA [50] and LLAMA [82] models, combining inpainting and prompt-based techniques for precise, contextually informed perturbations.

**Stress Testing Pipelines**    Stress testing pipelines, crucial in software engineering, remain under-utilized in machine learning. While various metrics exist for performance assessment and model comparison [67], they often lack the depth to fully capture model robustness and explain failure cases adequately. While initiatives like Stress Test NLI [56] focus on generating adversarial examples to evaluate models' inferential capabilities across six tasks, DynaBench [40] and CheckList [68] take a different approach by employing human-in-the-loop systems to dynamically benchmark and assess the robustness of natural language models in real-world scenarios. Simultaneously, in the vision community, Li et al. [46] utilize diffusion models to create ImageNet-E, honing in on assessing classifier robustness through object attributes, while Luo et al. [52] explore model sensitivity to user-defined text attributes using StyleGAN [2]. Building upon these endeavors, LANCE [62] advances the field by extracting insights from failures via a targeted perturbation algorithm, enabling stress testing across diverse attributes. Our work extends this paradigm to AI-generated image detection, presenting a versatile pipeline capable of performing image augmentation with varying magnitudes of perturbations across any diffusion model for a given set of image data points, facilitating evaluation and bias discovery in detector architectures through a comprehensive range of stress tests.

## 3 SEMI-TRUTHS

To precisely evaluate a detector's ability to distinguish between AI-generated and real images, we curate SEMI-TRUTHS, consisting of over $27,600$ real images and $1,329,155$ fake images. We

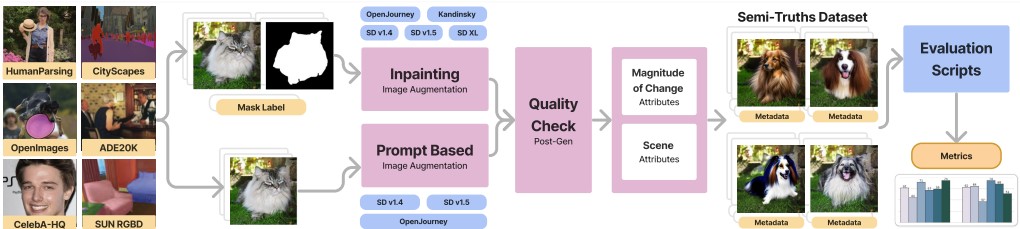

Figure 2: **End-to-end pipeline for SEMI-TRUTHS curation and detector stress testing.** The SEMI-TRUTHS pipeline sources data from 6 benchmarks and uses 2 perturbation techniques to perturb images. These images undergo saliency checks, metric computation, and stress testing of detectors across our curated tests based on the computed change metrics.

consider several crucial factors: (1) strategies for targeted augmentation at varying magnitudes, (2) diversification of scene distributions, (3) caption perturbation methods, (4) image augmentation techniques, and (5) the saliency of augmented images. The imbalance in our SEMI-TRUTHS dataset arises from pairing each real image with multiple augmented variations, a vital requirement for the benchmarking scheme as it enables a comprehensive exploration of model sensitivities across various dimensions (such magnitude of augmentation, change in subject matter, and augmentation technique). This section details methods to quantify augmentation magnitudes, followed by our generation and saliency check pipeline, and an overview of key dataset attributes.

| Small Changes | Medium Changes | Large Changes |
|---|---|---|
| Do not significantly alter the overall meaning or context of the image. This could include changing the color of a specific object, adding or removing a minor detail, adjusting the composition or perspective of the image, or slightly adjusting the color distribution of the image. | Slightly alter the viewer's perception of the image and its subject. They could involve minor changes to an object or its setting, like altering a background element, moving an object or person to another location within the frame, or changing the emotions of the people in the frame. | Involve substantial modifications to the image that fundamentally transform its interpretation or message. It may even appear surprising or strange to an audience. This could include altering, adding or removing major elements of the image background and making changes to the subject of the image. |

Table 2: **Semantic Taxonomy.** Definitions of the magnitudes of semantic change, used to guide the perturbation of image captions (for prompt-based-editing) and mask labels(for inpainting) using LLMs for targeted image perturbation.

## 3.1 Magnitudes of Augmentation

The alteration made to an image can be quantified in two ways: (1) the proportion of the image area that has been altered (*area ratio of change*), and (2) the degree to which the semantics of the image were altered (*semantic change*). To control the degree of alteration along these axioms, we start with an initial description of the image. This description is obtained by either selecting a segmentation mask and the corresponding class label or, in the absence of mask information, by generating a caption for the image using BLIP [45].

**Introducing Perturbations**   Motivated by the categorization of semantic and abstract content from visual semantics research [8], we create a taxonomy for small, medium, and large semantic changes (see Tab. 2). This taxonomy is used to guide the perturbation of an image caption or mask label using LLaVA-Mistral-7B [50] or LLAMA-7B [82] (see Sec. E). As shown in Fig.3, the model is provided with a semantic magnitude category, its definition, a caption to perturb, and the image (if using LLaVA-Mistral-7b). For prompt-based-editing, a diffusion model augments images based on perturbed captions, introducing semantic changes. In conditional inpainting, the perturbed mask label enables precise control over changes in the masked image region.

**Measuring Surface Area Change**   While segmentation masks help localize augmentations to an image, providing an area ratio of change, diffusion model imprecision can compromise their accuracy.

Dong et al. [17] demonstrate diffusion models can "color outside the box" during inpainting. Furthermore, the lack of mask guidance in prompt-based-editing necessitates the use of post-augmentation metrics to capture the size of alteration. Therefore we employ SSIM [89], MSE, and a custom metric that assesses the extent to which the structural components differ between the original and augmented images in pixel space. Our custom metric, derived from MSE, uses thresholding to remove noisy components followed by connected component analysis to generate masks indicating areas of change. Similar to the area ratio computed using the mask and the image, we compute a ratio using the generated mask to quantify the surface area of change. Each of these metrics are normalized between 0 and 1 and categorized into small, medium, and large changes based on percentiles: the bottom $25^{th}$ percentile for small, the $25^{th}$ to $75^{th}$ percentile for medium and anything beyond the $75^{th}$ percentile for large.

**Measuring Semantic Change** As mentioned previously, large language models (LLMs) are used to perturb image captions and mask labels with respect to the taxonomy of semantic change shown in Tab. 2. However, the stochasticity of LLMs and diffusion models necessitates the implementation of post-augmentation metrics that provide a quantitative measure of semantic change achieved. We use three different scores for this task: LPIPS [98], DreamSim [22] (both computed between the original and augmented images), and Sentence Similarity [78] (calculated between the original and perturbed captions/mask labels; see Sec. C.2). These metrics are normalized and categorized like Surface Area Change metrics, indicating small, medium, and large augmentations.

**Additional Metrics** In addition to surface area and semantic change, we incorporate metrics that provide a richer description of the underlying distribution, such as *scene diversity* and *scene complexity*. Scene diversity is defined by the number of unique elements within the original image, while scene complexity measures the quantity of each unique element. Both of these metrics are derived from instance segmentation maps (see Sec. C.2). Additionally, we distinguish changes by their spatial context - diffused changes are dispersed throughout the augmented image, whereas localized changes are concentrated in a specific area (see Algo. 2).

## 3.2 Image Augmentation Pipeline

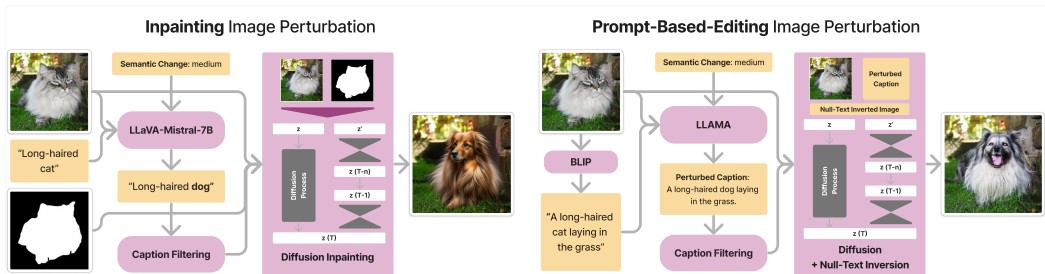

Figure 3: **Image Augmentation Pipeline.** Components of the image augmentation process for SEMI-TRUTHS curation using inpainting and prompt-based-editing methods.

Our image augmentation pipeline, delineated in Fig. 3, expands upon the work of LANCE [62] by integrating two distinct image augmentation techniques: (1) conditional inpainting and (2) prompt-based-editing. Both approaches leverage linguistic signal as guidance in image augmentation: prompt-based-editing requires a perturbation to the image caption using LLAMA-7B [82], and conditional inpainting relies on zero-shot mask label perturbation produced by LlaVA-Mistral-7B [50]. Furthermore, the complexity of this pipeline demands comprehensive saliency checks at various stages to ensure that augmented images maintain structural integrity and align with the specified directions of change. To this end, we implement two rounds of saliency evaluation within our image augmentation pipeline to identify instances of high-quality text and image augmentations.

**Caption Filtering** The first saliency check protocol evaluates LLM-perturbed captions to ensure two key aspects: (1) accuracy of generated BLIP [45] captions for prompt-based-editing in representing relevant image information, and (2) coherence and desirability of image edits produced

by perturbed captions/labels, ensuring semantic alignment with original content. For the former, CLIPScore [29] measures the difference between embeddings of the original image and its generated caption, filtering out the lowest 5th percentile values. For the latter, cosine similarity between CLIP [65] text embeddings of the perturbed caption/mask label and the original is calculated, removing values above the $95^{th}$ percentile (negligible change) and below the $5^{th}$ percentile (semantic incoherence). Additional details are mentioned in Sec. B

**Image Saliency Check** In the second stage of the saliency check pipeline, we aim to evaluate the (1) structural integrity of augmentations in the image while retaining resemblance to the original, and (2) semantic alignment of image augmentations with the perturbed captions/labels that were used to produce them. Since we lack reference images for direct comparison in image augmentations, conventional metrics like PSNR and SSIM are not suitable. Instead, we explored metrics that assess the structural integrity of each image individually. We use BRISQUE [53], a reference-free metric which quantifies the perceptual quality of an image, labeling images with a score under 70 as highly salient[2]. Similarly, we use CLIP similarity [65] between original and augmented images to ensure the diffusion model performed substantial enough augmentations on the original. We also employ CLIP directional similarity [23] to confirm that changes in images align with the changes in captions/labels. Images between the $20^{th}$ and $80^{th}$ percentile are considered highly salient. Additional details are mentioned in Sec. B

## 3.3 SEMI-TRUTHS Details

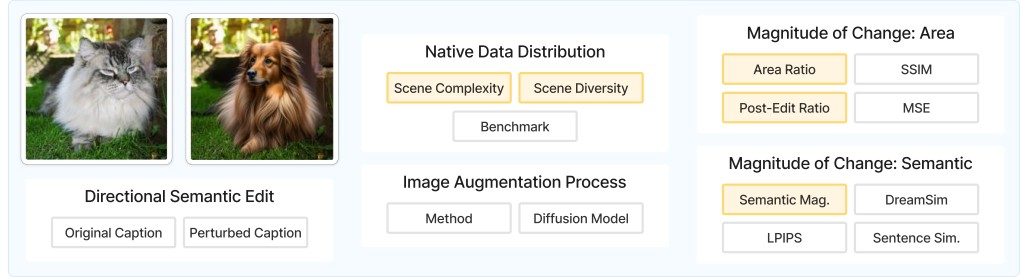

Figure 4: **SEMI-TRUTHS details and metadata.** Each augmented image in SEMI-TRUTHS is accompanied by metadata detailing properties related to the native data distribution, change magnitude (both area and semantics), and directional semantic edits. Attributes highlighted in yellow are novel contributions presented in this work.

**Data Distribution** We collect data from 6 semantic segmentation benchmarks representing various data distributions: CityScapes [12] for outdoor urban scenes, SUN RGBD [77] for indoor scenes, CelebA HQ [37] for human faces, Human Parsing [48] for full-body human images, and ADE20K [99] and OpenImages [43] for diverse themes. This combined dataset comprises $27,600$ real images and $223,400$ masks. Using conditional inpainting and prompt-based-editing techniques across 5 [61, 63, 74, 70] diffusion models for inpainting and 3 [63, 70] diffusion models for prompt-based-editing, with LlaVA-Mistral-7B [50] and LLAMA-7B [82] for prompt perturbation, we create $325,718$ prompt-based-editing datapoints and $1,003,437$ inpainting datapoints. After post-perturbation saliency checks, $\sim 74\%$ of images from inpainting and $\sim 55\%$ from prompt-based-editing techniques are labeled as highly salient, totaling $\sim 915,445$ images.

**Metadata** SEMI-TRUTHS encompasses extensive metadata accompanying both real and fake image pairs and masks, offering insights into every facet of the perturbation process (see Fig. 4). This metadata includes details about the source data distribution, such as the original benchmark from which the image was sourced, scene complexity and diversity (defined by the number and variety of scene elements), a list of unique entities present in each image, and the ratio of mask-occupied area. Additionally, it provides information about the diffusion model, perturbation technique, and language model utilized for each perturbation, alongside the original and perturbed caption/label. Furthermore,

---

[2]High BRISQUE scores are indicative of low perceptual quality.

each perturbed image is accompanied by quantitative and qualitative measures of change categorized across semantic and surface area-based metrics, as outlined in Sec. 3.1. The metadata also indicates whether the change is categorized as diffused or localized, determined using a custom algorithm (detailed in Algo. 2). All of this information is crucial for testing the effectiveness of detectors across various axes, as demonstrated in Sec. 4.

# 4 Experiments

| Detector | Backbone | Training Data Distribution | | | Precision(↑) | | | Recall(↑) | | |
|---|---|---|---|---|---|---|---|---|---|---|
| | | Scene | GANs | Diffusion | All | Real | Fake | All | Real | Fake |
| 1 DINOv2 [60] | ViT [18] + ResNet-50 [25] | General | ✗ | ✗ | 29.30 | 37.17 | 21.43 | 49.99 | 99.96 | 00.01 |
| 2 CNNSpot [86] | ResNet-50 [25] | General | ✓ | ✗ | 30.13 | 35.27 | 25.00 | 49.99 | 99.99 | 00.00 |
| 3 DIRE [88] | ResNet-50 [25] | General | ✗ | ✓ | 31.09 | 37.18 | 25.00 | 49.99 | 99.99 | 00.00 |
| 4 CrossEfficientViT [11] | EfficientNet-B0 [80] + ViT [18] | Face | ✓ | ✗ | 46.37 | 34.89 | 57.85 | 46.58 | 62.87 | 30.28 |
| 5 UniversalFakeDetect [59] | CLIP [65]-ViT [18] | General | ✓ | ✓ | 64.84 | 58.89 | 70.79 | 60.57 | 34.11 | 87.03 |
| 6 DE-FAKE [73] | CLIP [65] | General | ✓ | ✓ | 61.65 | 49.97 | 73.33 | 61.88 | 52.28 | 71.48 |

Table 3: **AI-generated Image Detectors evaluated with SEMI-TRUTHS**. We evaluated 6 detectors with varying backbones and training data distributions. Models that performed satisfactorily, highlighted in green, were selected for additional testing.

We conduct extensive experiments with SEMI-TRUTHS to evaluate the effectiveness of AI-generated image detectors in distinguishing real images from AI-augmented content (see Tab. 3). In the following sections, we show how knowledge abstraction over image augmentations in the dataset helps identify nuanced biases in various detectors. All evaluations are conducted on a 10% sample of SEMI-TRUTHS, totaling 87,000 images (27,000 real and 60,000 augmented). The evaluation dataset is available at: https://huggingface.co/datasets/semi-truths/Semi-Truths-Evalset. Since the real class (Real) is unaffected in the distribution-specific analysis, the key metric to observe is Recall on the augmented (Fake) class. A dip in Recall for a specific group indicates the detector's sensitivity to that augmentation. Detector default settings (provided in their respective codebases) have been used for conducting evaluations.

**Overall Detector Performance** We select a diverse set of open-source AI-generated image detectors for stress testing. As demonstrated in Tab. 3, each model has a unique architecture and training distribution. We first evaluate these detectors in a zero-shot setting using metrics like Precision, Recall, and F1-Score to identify top performers for further analysis. Of the 6 models [3] chosen, only half demonstrated adequate performance for continued evaluation. The underperforming models include (1) DinoV2, a foundation vision model leveraged for zero-shot AI-generated image prediction, (2) CNNSpot, a ResNet-50 trained solely on GAN-generated content, and (3) DIRE, a ResNet-50 trained on diffusion-generated content.

**Sensitivity to Data Distribution** To assess potential biases toward specific data distributions, we inspect detector performances on various semantic segmentation benchmarks represented in SEMI-TRUTHS. Fig. 5 shows that detector performance varies significantly across data sources. Notably, CrossEfficientViT [11], which is trained on GAN-generated images of human faces, exhibits a significant performance drop on human faces sourced from benchmarks ADE20K, CityScapes [12], and SUN-RGBD [77] (CrossEfficientViT pre-emptively filters any images that do not contain a human face). In contrast, DE-FAKE [73], trained on general scene images, exhibits the worst performance on CelebA-HQ [44] and HumanParsing [48] due to limited focus on humans and portrait-like images in its training distribution. On the other hand, UniversalFakeDetect [59], trained on indoor bedroom images and other generic scenes, fails to perform well with SUN RGBD and shows a significant performance drop on CityScapes.

Furthermore, we investigate the detectors' ability to handle highly complex and diverse multi-instance scenes. Their performance is evaluated across varying degrees of scene diversity (number of unique class instances in the images) and scene complexity (number of instances in total), categorized into small, medium, and large bins (additional details in Sec. C.2). We find that UniversalFakeDetect's [73] performance drops with increasing scene diversity and complexity. In contrast, DE-FAKE [73]

---

[3] These models represent a range of state-of-the-art AI image detectors, showcasing SEMI-TRUTHS's versatility. The evaluation pipeline enables easy benchmarking of new detectors across standardized tests. See: https://github.com/J-Kruk/SemiTruths.

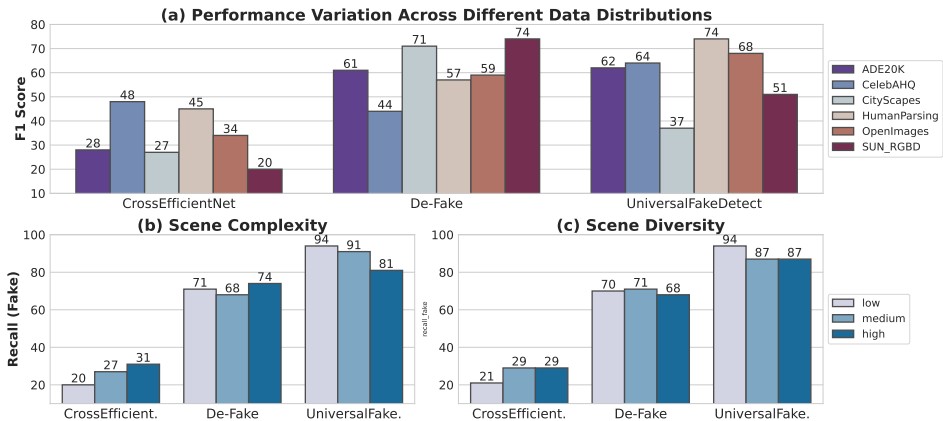

Figure 5: **Detectors are sensitive to semantic aspects of data distribution.** The 3 detectors, CrossEfficientViT, DE-FAKE and UniversalFakeDetect were evaluated across varying (a) data distribution, (b) scene complexity and (c) scene diversity.

remains fairly robust across different scene variations. CrossEfficientViT [11] shows improved performance with increasing scene complexity and diversity, which can be attributed to human-centered benchmarks like CelebA-HQ [44] and HumanParsing [48] segmenting distinct facial features and body parts which results in lower scene complexity.

These results highlight that detectors are highly sensitive to the semantic attributes of data distributions, emphasizing the importance of stress tests to identify and address distributional weaknesses.

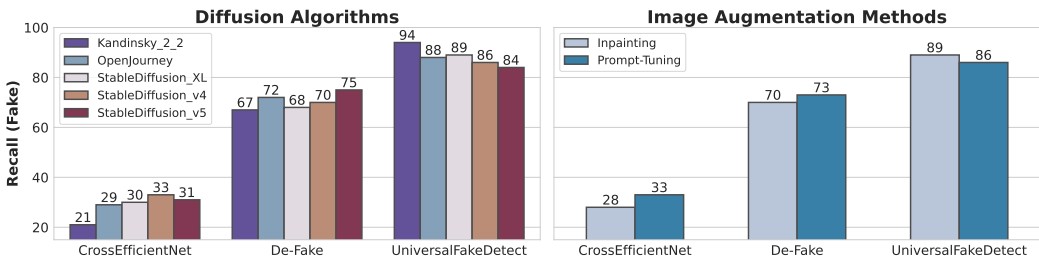

Figure 6: **Performance variation across image augmentation methods and diffusion algorithms.** SEMI-TRUTHS offers data generated using various diffusion algorithms and augmentation methods facilitating detector evaluation on these aspects.

**Evaluation across Augmentation Techniques and Models**  SEMI-TRUTHS contains images generated using two different augmentation approaches - conditional inpainting and prompt-based-editing - as well as five different diffusion algorithms: StableDiffusion v1.4, StableDiffusion v1.5, StableDiffusion XL [61], OpenJourney [63], and Kandinsky 2.2 [74]. This diversity in generated content enables investigation of detector sensitivities to different augmentation procedures.[4] As shown in Fig.6, UniversalFakeDetect [59] performs best on images augmented with Kandinsky 2.2 [74] and worst on those augmented with StableDiffusion v1.5 [70], with a 10% difference in Recall score. The inverse is true for DE-FAKE [73]. CrossEfficientVit [11] performs best on images augmented with StableDiffusion v1.4 and worst with Kandinsky 2.2 [74], with a 12% drop in performance. Furthermore, CrossEfficientViT [11] and DE-FAKE [73] are more sensitive to inpainted images, whereas UniversalFakeDetect [59] performs worst on content augmented with prompt-based-editing.

**Evaluation across Varying Magnitudes of Augmentation**  As detailed in Sec. 3.1, each image in SEMI-TRUTHS is fitted with an array of descriptive attributes that capture the magnitude of change. In Fig.7 we examine the impact of varying degrees of augmentation on detector performance, focusing on both surface area and semantic changes. Note that CrossEfficientViT [11] performs better on smaller values of Area Ratio, where as UniversalFakeDetect [59] performs better on larger changes.

---

[4]Limitations of  [28],  [54] restrict prompt-based-editing to OpenJourney, StableDiffusion v1.4 & v1.5

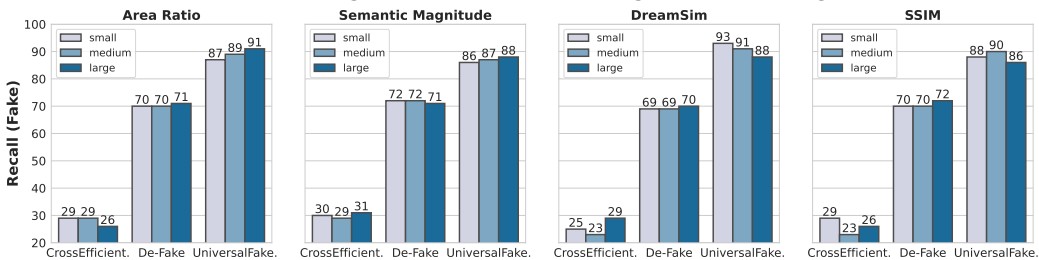

Figure 7: **Performance variation of select detectors across various magnitudes of augmentation.** DE-FAKE [73] is robust across the board, Area Ratio captures the sensitivity exhibited in UniversalFakeDetect [59] and CrossEfficientViT [11].

| Phrase(Original ⟶ Edited) | Counts | Recall |
|---|---|---|
| Easy cases | | |
| 1 lower lip ⟶ nose | 70 | 66.67 |
| 2 left brow ⟶ left brow with slight arch | 99 | 50.0 |
| 3 car ⟶ car with shiny chrome accents | 59 | 45.16 |
| Difficult cases | | |
| 4 lower lip ⟶ lipstick | 190 | 15.79 |
| 5 skin ⟶ skin with subtle freckles | 127 | 7.14 |
| 6 left ear ⟶ earring | 177 | 6.67 |

(a) **CrossEfficientViT [11]**

| Phrase(Original ⟶ Edited) | Counts | Recall |
|---|---|---|
| Easy cases | | |
| 1 skin ⟶ leather | 74 | 98.65 |
| 2 nose ⟶ nose ring | 138 | 97.1 |
| 3 left ear ⟶ earring | 177 | 96.61 |
| Difficult cases | | |
| 4 vegetation ⟶ tree | 225 | 66.67 |
| 5 ego vehicle ⟶ mercedezbenz | 161 | 65.84 |
| 6 vegetation ⟶ building | 150 | 65.33 |

(b) **UniversalFakeDetector [59]**

| Phrase(Original ⟶ Edited) | Counts | Recall |
|---|---|---|
| Easy cases | | |
| 1 car ⟶ car with shiny silver paint | 57 | 85.96 |
| 2 vegetation ⟶ tree | 225 | 84.89 |
| 3 ego vehicle ⟶ mercedezbenz | 161 | 81.37 |
| Difficult cases | | |
| 4 skin ⟶ skin with subtle freckles | 127 | 62.99 |
| 5 nose ⟶ nose ring | 138 | 58.57 |
| 6 skin ⟶ leather | 74 | 58.11 |

(c) **De-FAKE [73]**

Table 4: **Directional Semantic Edits for investigating detector biases.** Directional Semantic Edits provide insights on which edits to a certain entity has a higher chance of fooling detectors, we notice that patterns vary significantly across detectors.

UniversalFakeDetect's [59] performance also drops as DreamSim [22] scores increase. Even though DE-FAKE [73] is not the best performing model, it appears to be the most robust against various magnitudes of change across the board.

**Directional Semantic Edits**  Quantitative metrics can be reductive when describing how the semantics or narrative of an image changes. Transitioning to an embedded space to assess similarity, for example, often results in significant information loss. To address this issue, we introduce "Directional Semantic Edits", which groups augmented images from SEMI-TRUTHS by distinct pairs of original and perturbed caption/mask labels. In the evaluation set, certain directional semantic edits occurred as frequently as 445 times. Each detector is evaluated on these groups, and metrics are sorted by Recall, as shown in Tab. 4. Each model exhibits distinct performance variations based on specific semantic changes. Notably, UniversalFakeDetect [59] performs best on augmentations to facial features but worst on augmentations to vegetation. Conversely, DE-FAKE [73] excels at detecting augmentations to cars and vegetation but struggles with augmentations to human faces. CrossEfficientViT [11] shows varied performance with augmentations to human faces, appearing in both its highest and lowest ranks, indicating sensitivity to the magnitude of the change.

Further analysis of these augmentations can maximize the potential of these algorithms by informing decisions about the most suitable ensemble techniques. For example, while UniversalFakeDetect [59] struggles with vegetation-to-tree augmentations, DE-FAKE [73] excels, suggesting a suitable combination for ensemble approaches. Such analysis reveals the most challenging directional augmentations, offering insights into detector model limitations.

**Surveying Human Perception of Magnitudes of Change**  To build intuition about the algorithms we use to quantify the degree of visual and semantic change achieved during image augmentation, we conduct a user study to evaluate if any metrics align with human perception. Annotators are asked to categorize changes between original and augmented images as "not much," "some," or "a lot," corresponding to our "small," "medium," and "large" change bins. We then compute correlation coefficients (Pearson [41], Kendall Tau [64], and Spearman [1]) between human scores and quantitative measures in SEMI-TRUTHS. The results in Tab. 5 show that Area Ratio, a novel metric presented in this work, demonstrates the highest correlation with human perception, whereas other metrics demonstrate little to no correlation. It is important to note, however, that some changes may be imperceptible to the human eye but appear drastic in pixel space (additional discussion in Sec. D).

| Correlation Coeff. | Change Metrics(↑) | | |
|---|---|---|---|
| | Area Ratio | LPIPS Score | SSIM |
| 1 Pearson | 0.46 | 0.14 | −0.16 |
| 2 Kendall-Tau | 0.40 | 0.15 | −0.14 |
| 3 Spearman | 0.50 | 0.19 | −0.17 |

Table 5: **Correlation between quantitative measures of change and Human Perception.** Correlation coefficients computed between human-annotated magnitudes of change and quantitative metrics available in the dataset. Quantitative metrics not displayed here had coefficients < 0.10.

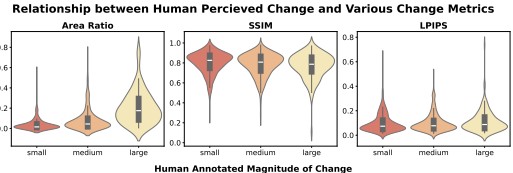

Figure 8: **Relationship between quantitative change metrics and Human Perception of change (small, medium, large) in** SEMI-TRUTHS. Each violin plot shows the distribution of metric values for a change category.

## 5 Discussion

**Limitations and Future Work**  Our inpainting pipeline currently relies on manual semantic mask input from existing semantic segmentation benchmarks, limiting usability. To improve, automatic mask generation methods like SAM [42] can be embedded into the augmentation pipeline, similar to InstructEdit [85]. Additionally, using LLAMA-7B [82] and LlaVA [50] models for zero-shot perturbation has led to many poor-quality outputs, requiring filtering. Future iterations will involve fine-tuning these models. We are also aware of potential biases in metrics like LPIPS [98], Sentence Similarity [78], and DreamSim [22], which may impact evaluations. To mitigate this issue, we will incorporate a combination of multiple open-source LLMs to compute semantic change metrics, thereby reducing the inherent biases associated with any single model, a process facilitated by our modular pipeline which enables easy switching between different LLMs of the user's choosing.

**Ethical Issues and Bias Mitigation**  While our project aims to create a test suite for evaluating and improving detector robustness, it can also be used to create fake images capable of deceiving AI-generated image detectors, potentially facilitating the spread of misinformation. Hence, we curated SEMI-TRUTHS by sourcing images from publicly available datasets with minimal potential for harm, and any manipulations on such images should not serve as a potential threat to society as per our knowledge. Additionally, despite our efforts at diversification of data and models, inherent biases from these modules may persist, potentially perpetuating or exacerbating existing inequalities, resulting in uneven performance across different contexts and types of images. To minimize additional bias, we employ a diverse range of perturbation techniques, diffusion models, LLMs, and source benchmarks. This diversity, along with comprehensive metadata—including original and perturbed captions/labels and images—enables users to analyze perturbation styles and identify existing biases in the generative models.

## 6 Conclusion

To address the rising threat of misinformation from AI-augmented images, we introduce SEMI-TRUTHS: a comprehensive dataset of $1,329,155$ AI-augmented images, with detailed metadata on source distribution, augmentation techniques, change magnitudes, diffusion models, and directional edits (paired original and perturbed captions). Our plug-and-play image perturbation pipeline enables easy generation of additional augmentations and offers a standardized platform to test detector robustness across curated scenarios. Our analysis reveals that state-of-the-art detectors exhibit varying sensitivity to perturbation levels, data distributions, and augmentation methods, providing valuable insights into detector functionality. With a semantic taxonomy for defining change types and a quality-check pipeline, SEMI-TRUTHS also serves as a robust training and testing resource, enhancing the resilience of AI-generated image detectors. Furthermore, its diverse metadata enables bias analysis, supporting research into model fairness. We believe the user-friendly design of SEMI-TRUTHS will facilitate ongoing research into robustness against evolving generative models, helping combat misinformation effectively.

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

# A    Image Augmentation

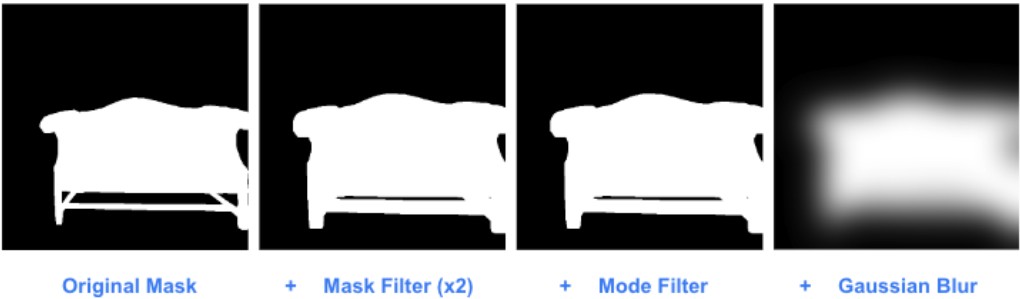

Figure 9: This figure demonstrates how each mask is preprocessed before it is used to augment images through diffusion inpainting. Two Max Filters are applied with width 9, following a mode filter to smooth edges, and finally Gaussian blur.

## A.1    Data Preprocessing

We preprocess and condense all benchmarks into one by standardizing segmentation masks, sizes of visual media, and metadata. Area ratios are computed for every mask, this is a value that denotes the percentage of the original image highlighted by the segmentation. Masks that are too small, i.e. denote a region that is less than 5% of the image area, are excluded. We process all masks by applying two Max Filters (size 9), one Mode Filter (size 9) to smooth edges, followed by Gaussian blur (radius 16), as demonstrated by Fig.9. This procedure is optimized by inspecting the quality of image augmentations produced. Fig.10 highlights how diffusion inpainting outputs are affected by various magnitudes of Gaussian blur and Max filtering.

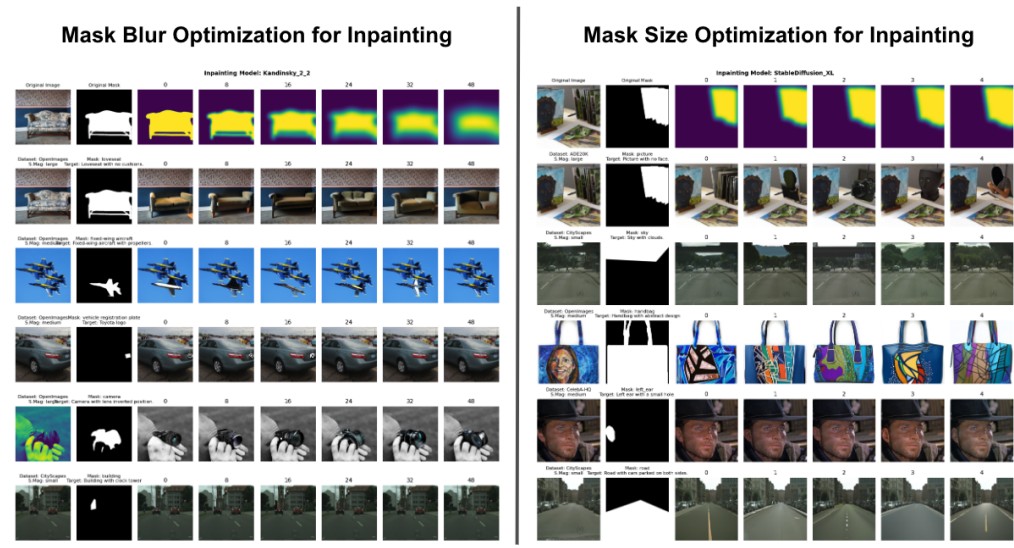

Figure 10: This figure demonstrates how each mask is preprocessed before image augmentation through diffusion inpainting. Two Max Filters are applied with width 9, a Mode Filter to smooth edges, followed by Gaussian blur.

## A.2 Visualizing Capabilities of Various Diffusion Models

To select the diffusion models used for image augmentation in SEMI-TRUTHS, a qualitative investigation is conducted to evaluate the quality of the generated content. Fig.11 displays one such exploration, in which we compare the inpainting capabilities of each diffusion model on a number of image, mask pairs.

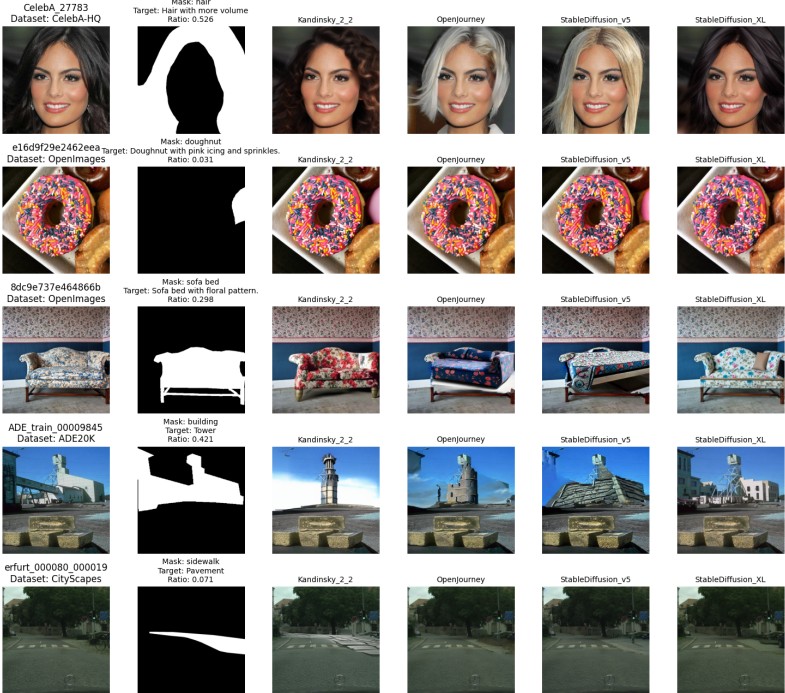

Figure 11: Exploring capabilities of diffusion models for inpainting image augmentation.

# B Quality Verification

## B.1 Caption Filtering

The first step in ensuring high quality generations is to select high quality prompts that will be provided as to diffusion models. In addition to the steps mentioned in the paper, we also filter the generated captions based on the following criteria:

1. Length of perturbed text: To ensure high quality generation, we prune all generated text. In the case of inpainting, we discard any perturbed label larger than 5 words. For prompt-tuning we discard any perturbed captions larger than 30 words.

2. Special characters: We observe that models used for image augmentation do not respond well to the inclusion of special characters in the prompts. Hence, any caption or label containing a special character is discarded.

## B.2 Image Saliency Check

Additional details regarding quality check metrics mentioned in Sec.3.2, as well as metrics used after the image perturbing phase to determine the quality of images, are given below:

1. *Directional Similarity score:* This metric ensures the change in the images is reflective of the change in the captions/label, and the generated image aligns with the perturbed caption/label. Directional similarity is calculated as follows:

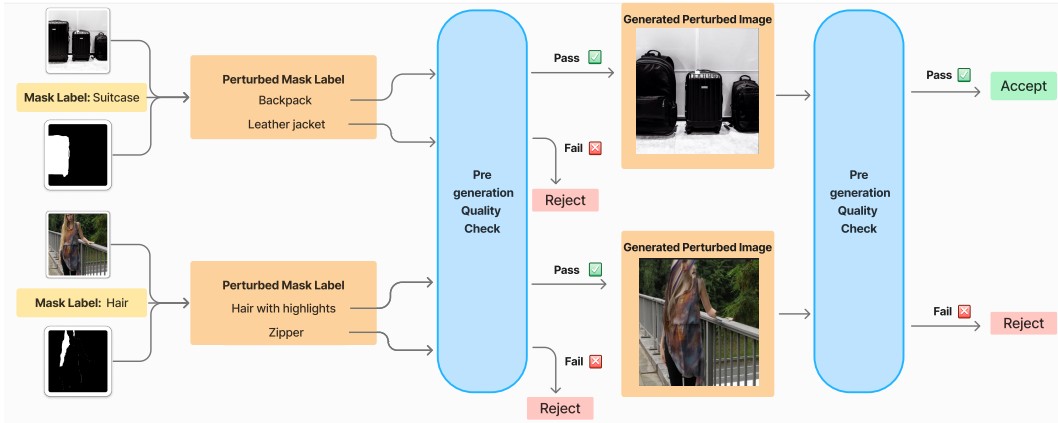

Figure 12: Overview of the quality check pipeline: Low-quality captions or label perturbations are rejected before image generation, and low-quality images are rejected at the end of the pipeline. Only generations that maintain high quality throughout the process are accepted into the final dataset.

$$Cosine(CLIP_{\text{original image}} - CLIP_{\text{perturbed image}},$$
$$CLIP_{\text{original text}} - CLIP_{\text{perturbed text}})$$

We retain the images that correspond to the values lying above the $23^{rd}$ percentile of the distribution.

2. *Remove completely black images:* In certain cases, if the prompt is complex or considered potentially harmful, the diffusion models mentioned in Sec. 3.3 generate completely black images with no information. We remove all such generations from the dataset.

Examples of the images corresponding to varying distribution of metrics are shown in Fig.13.

Through our quality check process, we assign a True or False label depending on whether they pass the saliency check. For an image to pass the saliency check it must simultaneously satisfy all

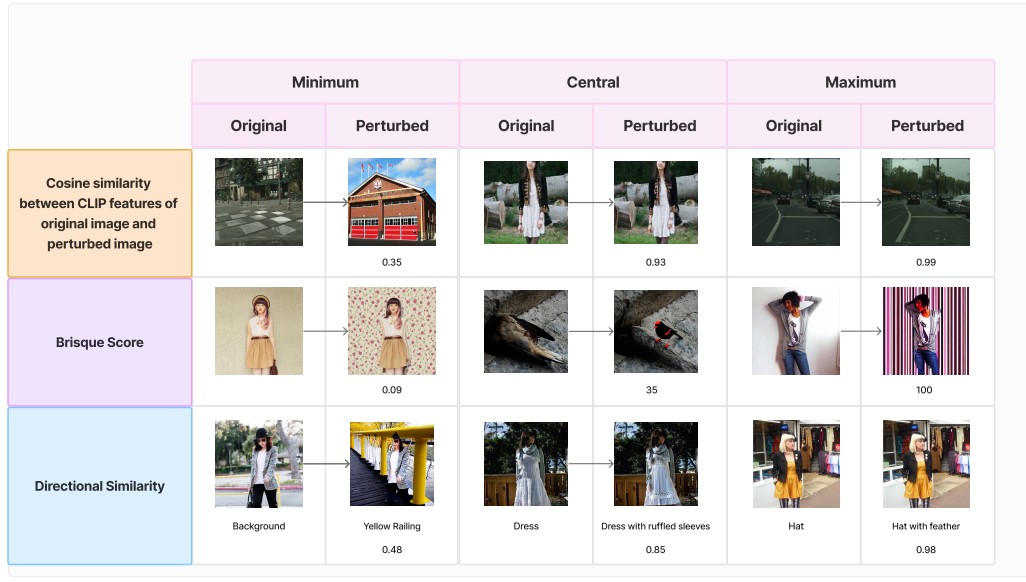

Figure 13: Examples of original and perturbed images throughout the spectrum of each quality check metric, from minimum to maximum

the following conditions: (1) have a Brisque score below 70 (2) have the CLIP image similarity the original and augmented image lie between the 20th an 80th percentile values, and (3) have the directional similarity score lie above the 23rd percentile value. Examples of filtered out images are shown in **??**

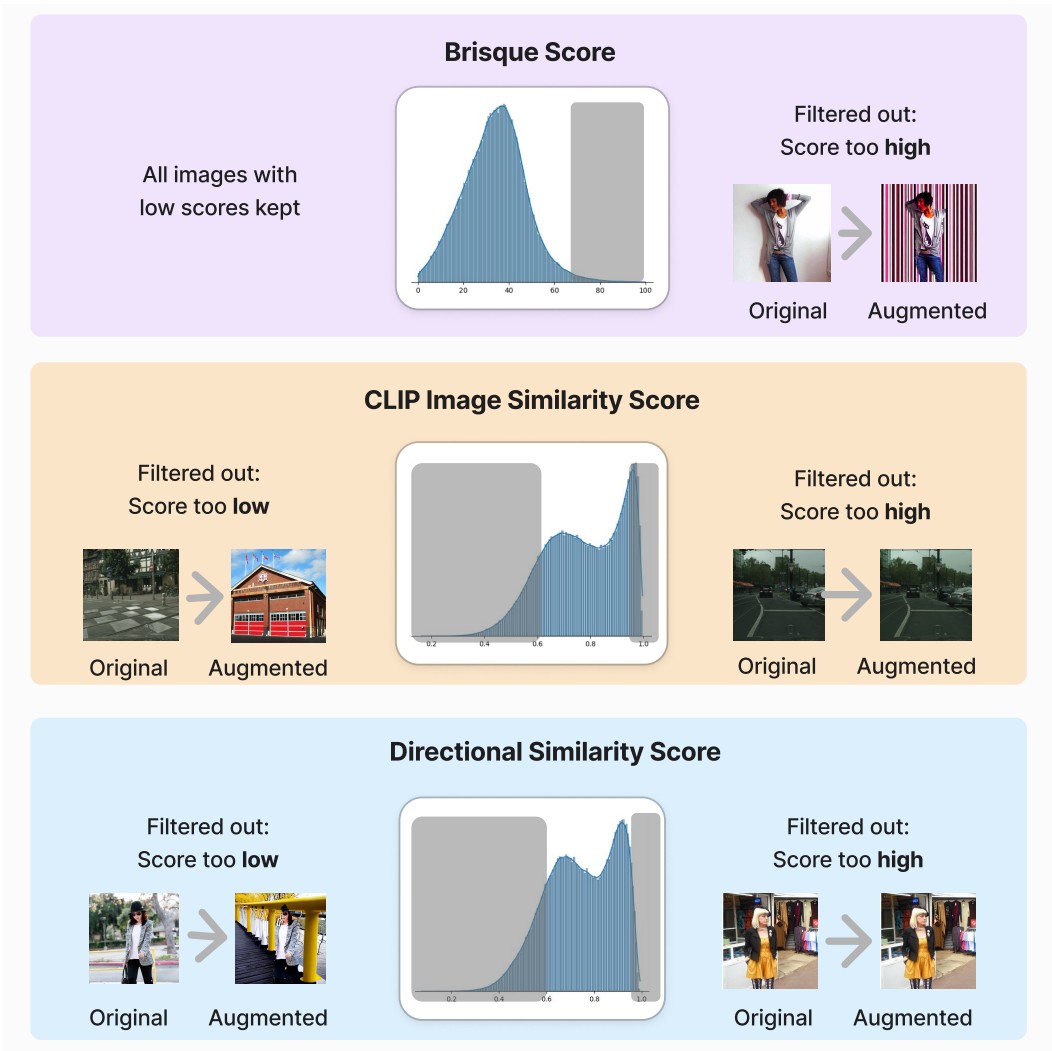

Figure 14: Distribution of quality check metrics across prompt based perturbations. The images lying in the grayed out areas are discarded during quality check.

## C    Experimental Design

### C.1    Cropping of Detector Input Images

Most AI-generated image detection models evaluated in our experiments prepare input images by resizing and taking a 224x224 center-crop. For our experimental setup, we acknowledge that this method of pre-processing may lead to cases where the perturbations made to the image lie outside the cropped area. To address this concern, we implemented a method of cropping the input image in a way that does not exclude the augmentation using the image perturbation masks. For inpainting the generation masks are used, and for prompt tuning we use the masks generated after image perturbing.

The part of the image to be perturbed is masked using white pixels. This original mask is center-cropped, and the surface area of white pixels in the cropped mask is compared with the surface area of white pixels in the original mask. If this surface area ratio passes a specified threshold, it is

---

**Algorithm 1** Detector Input Image Cropping Method

---

1: **Input:** input image $img$, perturbation mask $mask$, initial transformation involving center crop $T$, threshold for good crop $thresh$, modified transformation $T'$
2: **Initialize:** $goodCrop \leftarrow False$
3: **Initialize:** $centerPixel \leftarrow (maskHeight/2, maskWidth/2)$
4: **while** $goodCrop \neq True$ **do**
5:      $maskTrans \leftarrow T(mask)$
6:      $maskSA \leftarrow$ numWhitePixels($mask$)
7:      $maskTransSA \leftarrow$ numWhitePixels($maskTrans$)
8:      $maskRatio = maskSA/maskTransSA$
9:      **if** $maskRatio \geq thresh$ **then**      ▷ Mask surface area preserved after resize and cropping
10:          $goodCrop \leftarrow True$
11:          $centerPixel \leftarrow$ updateCenterPixel()      ▷ Save new center pixel to crop input image
12:      **else**
13:          $centerPixel \leftarrow$ center(randomCrop($mask$))   ▷ Recrop randomly and save center pixel
14:          **continue**
15:      **end if**
16: **end while**
17: $T' \leftarrow$ transform($centerPixel$)      ▷ Use the new center pixel for custom cropping
18: $croppedImage \leftarrow T'(img)$
19: **Return:** $croppedImage$      ▷ Final Cropped Image Input

---

determined that the cropped area includes a sufficient amount of the perturbed area for the detectors to identify, and this transformed input is passed in through the detector.

## C.2 Metric Binning Strategy

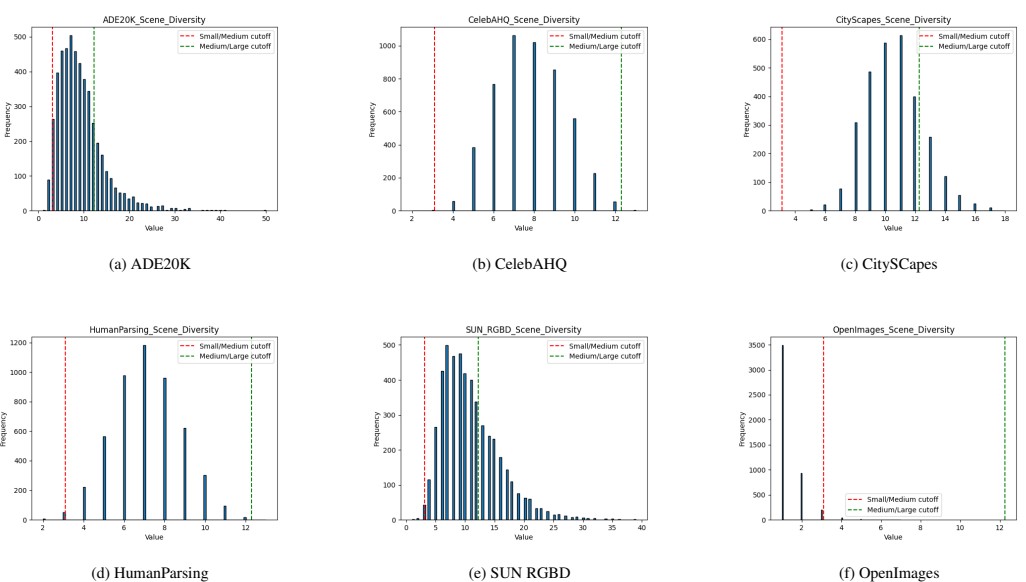

(a) ADE20K      (b) CelebAHQ      (c) CitySCapes

(d) HumanParsing      (e) SUN RGBD      (f) OpenImages

Figure 15: Scene diversity visualized across the datasets with cutoffs for binning

**Scene Diversity**     We use scene diversity of the original image to determine how the diversity of the scene plays a role in misleading AI-generated image detectors. To calculate scene diversity, we use class labels provided by the segmentation masks to understand the composition of the scene by finding how many unique classes are present in each scene. We then divide the values into three bins of small, medium and large where the values below 1 standard deviation from the mean are assigned the bin small, the values above 1 standard deviation from the mean are assigned the bin large and the

rest are assigned the bin medium. Fig. 15 illustrates the scene diversity distributions for the entire source dataset as well as for each of the individual source datasets.

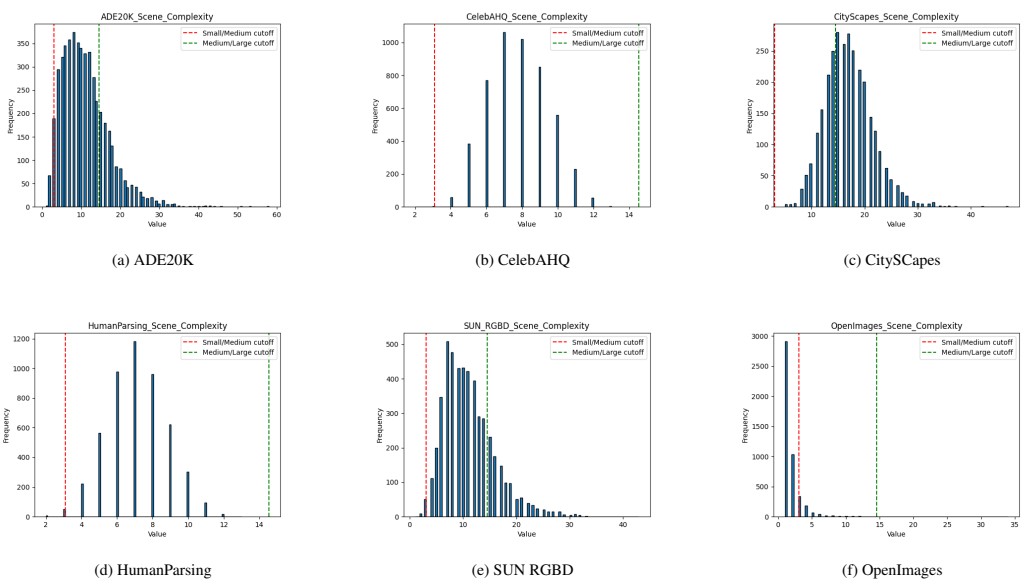

Figure 16: Scene complexity visualized across the datasets with cutoffs for binning

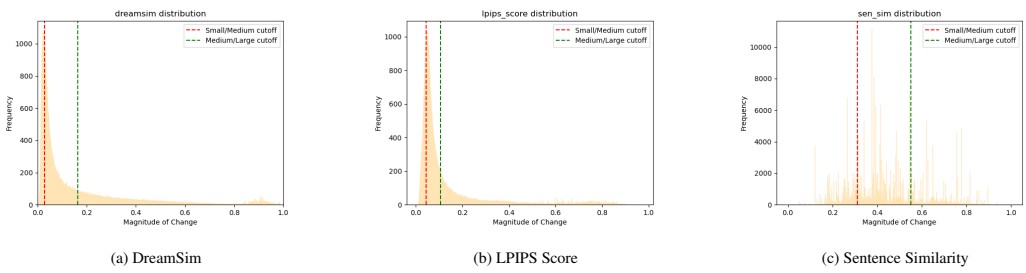

Figure 17: Distribution across different semantic change metrics with their binning cutoffs

**Scene Complexity** We use scene complexity of the original image to determine how the complexity of the scene plays a role in misleading AI-generated image detectors. To calculate scene complexity, we count the number of instances present in each image using the provided instance segmentation masks. We then divide the values into three bins of small, medium and large where the values below 1 standard deviation from the mean are assigned the bin small, the values above 1 standard deviation from the mean are assigned the bin large and the rest are assigned the bin medium. Fig. 16 illustrates the scene complexity distributions for the entire source dataset as well as for each of the individual source datasets.

**Semantic and Surface Change Metrics** For semantic and surface area change, these values are categorized into 3 bins (small, medium and large), where the bottom $25^{th}$ percentile was assigned to small changes, the middle $25^{th}$ to $75^{th}$ percentile for medium and above $75^{th}$ for large. The distribution of changes across the different bins for the entire dataset along with the cutoffs for binning is illustrated in Fig. 17.

### C.3 Post Perturbation Surface Area Change Metric

The Algo. 2 indicates the generation of post-perturbation surface area change metric and corresponding mask. We also use these metrics to calculate additional metrics that categorize a change as

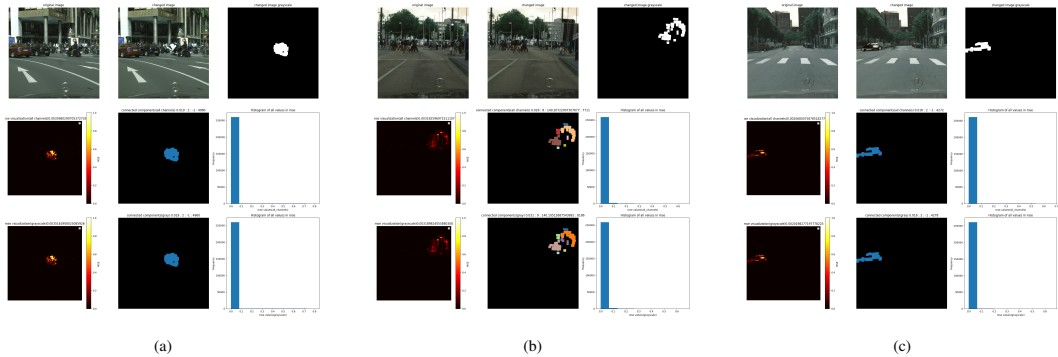

Figure 18: **Examples of localized changes** Steps in the calculation of post-perturbation mask and corresponding localized change identification

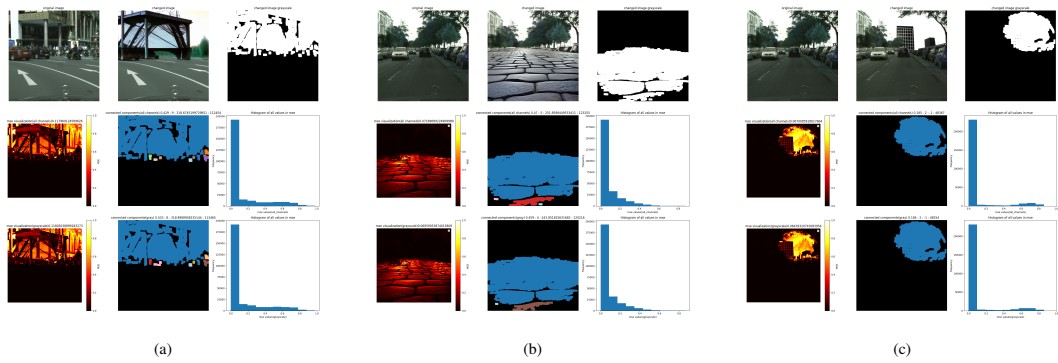

Figure 19: **Examples of diffused changes** Steps in the calculation of post-perturbation mask and corresponding diffused change identification

localized or diffused. Here localized means that the changes are concentrated in one area whereas diffused indicates that the change is spread out throughout the image. The algorithm also highlights how we use information gathered from the connected components to classify a change as diffused or localized. Fig. 18 and Fig. 19 illustrate a few examples of diffused and localized changes.

## D    Human Evaluation

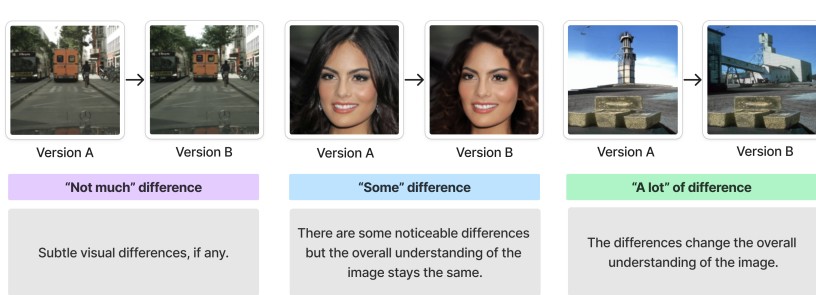

Figure 20: The instructive figure provided to annotators during the user study. The visualization was design to guide the participant on what kind of inputs are expected, without priming them on explicit definitions that we used in designing our semantic change categories.

---

**Algorithm 2** Post-Perturbation Surface Area Change ($I_1, I_2$)

---

1: Initialize key parameters
2: **Input:** $I_1$ (original image), $I_2$ (perturbed image)
3: **Threshold:** $T \leftarrow 0.1$
4: **Kernel:** $K \leftarrow$ white_pixels$(11, 11)$
5: # Calculating mean squared error between the Real and Perturbed RGB images
6: $mse_{rgb} \leftarrow (I_1[:,:,0] - I_2[:,:,0])^2 + (I_1[:,:,1] - I_2[:,:,1])^2 + (I_1[:,:,2] - I_2[:,:,2])^2$
7: # Normalization to (0,1)
8: $mse_{rgb}^{norm} \leftarrow \frac{mse_{rgb}}{(255 \times 255 \times 3)}$
9: # Thresholding to remove noise
10: $mse\_thresh \leftarrow$ Remove_Noise$(mse_{rgb}^{norm}, T)$
11: # Generate binary mask
12: $binary\_mask \leftarrow mse\_thresh > 0$
13: # Dilate mask
14: $dilated\_mask \leftarrow$ Dilate$(binary\_mask, K)$
15: # Find all connected components for the mask
16: $CC, CC\_img, Stats, CC\_centroids \leftarrow$ Get_connected_components$(dilated\_mask)$
17: # Merge neighbouring components
18: $merged\_components \leftarrow$ Merge$(CC\_img, CC\_centroids)$
19: # Remove extremely small components to get final post perturbation mask
20: $post\_edit\_mask \leftarrow$ Remove_Small$(merged\_components, \text{min\_size}, \text{min\_connected})$
21: # Calculate post perturbation ratio
22: $post\_edit\_ratio \leftarrow \frac{\text{num\_white\_pixels}(post\_edit\_mask)}{\text{total\_pixels}(post\_edit\_mask)}$
23: # Calculate the largest distance between the connected components
24: $max\_centroid\_dist \leftarrow \max(\text{pdist}(CC\_centroids))$
25: # Get the ratio of the largest connected component
26: $largest\_CC\_ratio \leftarrow \frac{\max(\text{calc\_CC\_size}(Stats))}{\text{total\_pixels}(post\_edit\_mask)}$
27: # Categorize as localized or diffused change
28: **if** $largest\_CC\_ratio \geq 0.2$ **or** (#CC > mean(#CC) **and** max_centroid_dist) **then**
29:     change $\leftarrow$ Diffused
30: **else**
31:     change $\leftarrow$ Localized
32: **end if**

---

The annotated dataset contains 800 image pairs, 100 for each augmentation method and diffusion model combination. A total of 145 individuals contributed to this survey through Amazon Mechanical Turk, with an approximate compensation of $ 15 per hour. Quality was maintained through 25 pre-annotated image pairs that served as attention checks, as well as qualifications that were granted to a community of vetted annotators. Three unique annotations were collected per image pair, for which the Interclass Correlation (ICC2k) score across all image pairs is 0.835. The final human perception score was the mode of the set.

The motivation behind this experiment is to evaluate how various means of quantifying magnitude of change in an image relate to human perception. The results show that the Area Ratio metric, a contribution of this work, had the highest correlation with human annotation. However, its important to note that specific metrics, such as MSE RGB, are incredibly sensitive to the slightest of changes. Whereas high-performing generative models can alter images in a way that would not be perceived by the human eye. Therefore, some metrics used in SEMI-TRUTHS capture important information about image augmentation even if they do not correlate with human perception.

This user study was IRB Exempt, as it did not obtain/access any private, personally identifiable, or demographic data about the participants. The only information provided was a statement on perceived change between two images.

# E  Perturbation Model Experiments

We considered different types of models to perform the perturbations in both the generated captions, and the mask label. We used both unimodal (LlaMA 2) and multimodal (LLaVA Mistral 7B, LLaVA Hermes 34B) models to perform these perturbations and compared their outputs to settle on our final pipeline.

We find that the performance of these models varied significantly across different perturbation methods. Initially, we used multimodal models to suggest perturbations, as they could process images as input and thereby capture contextual information that might be missed when only captions are provided. However, we observed that the LlaVA Mistral 7B model performed poorly in caption perturbation in the prompt-perturbation method. To address this, we scaled up the model size by incorporating LlaVA Hermes 34B into our pipeline. Surprisingly, we discovered that despite scaling up the best caption perturbations for prompt-perturbation were achieved using the LlaMA model instead. A comparison of the perturbed captions and the related generated images can be found in Fig.21. However we maintain LlaVA Mistral 7B as part of the inpainting pipeline as it shows best performance and also retains image context during suggesting perturbed labels for masks.

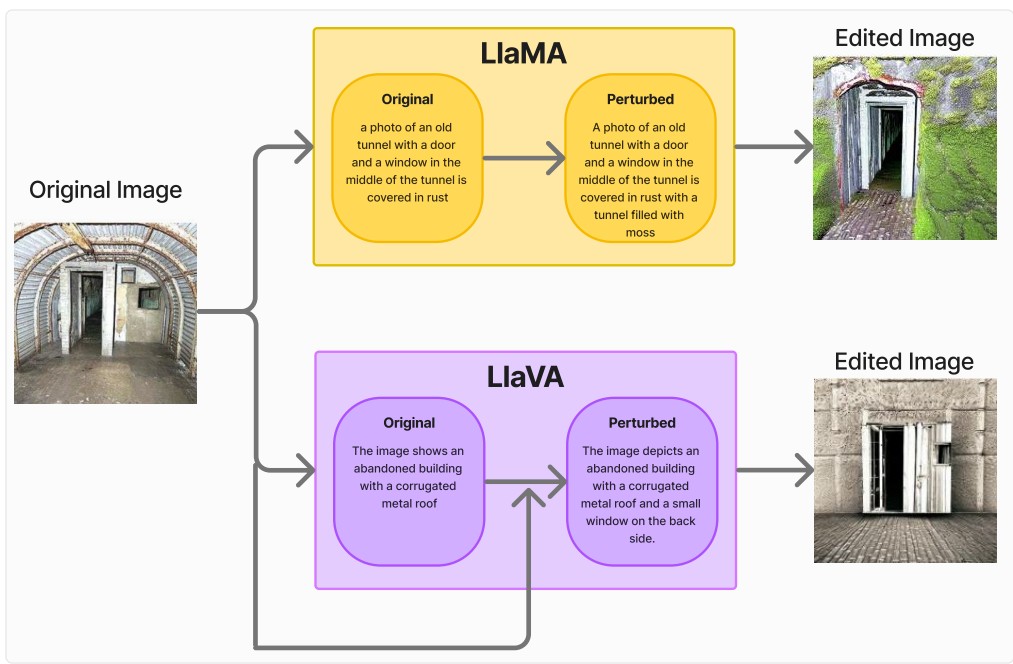

Figure 21: Examples of perturbations made by LlaMA and LlaVA, and their resulting images.

# F  Evaluation Metrics

The choice of metrics plays an essential role when evaluating on an imbalanced dataset. Therefore, we report individual class Precision, Recall and F1, along with overall Precision, Recall and F1, to provide a more interpretable overview. The majority of our experiments emphasize Recall on the fake class, as the impact of various augmentations primarily affects this class while leaving real images largely unaffected. However, we recognize that additional metrics addressing class imbalance would offer a more comprehensive evaluation. Consequently, we have included AUC-ROC and AUC-PR curves for both the original and a balanced evaluation set (containing 27,000 real and fake images) for the experiments mentioned in Tab. 3.

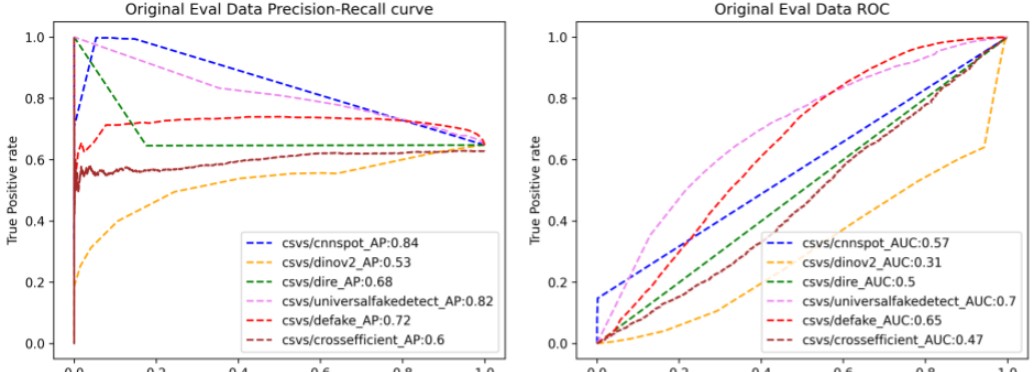

Figure 22: Precision-Recall curve with AP and AUC-ROC for each detector evaluated on Semi-Truths Eval Dataset

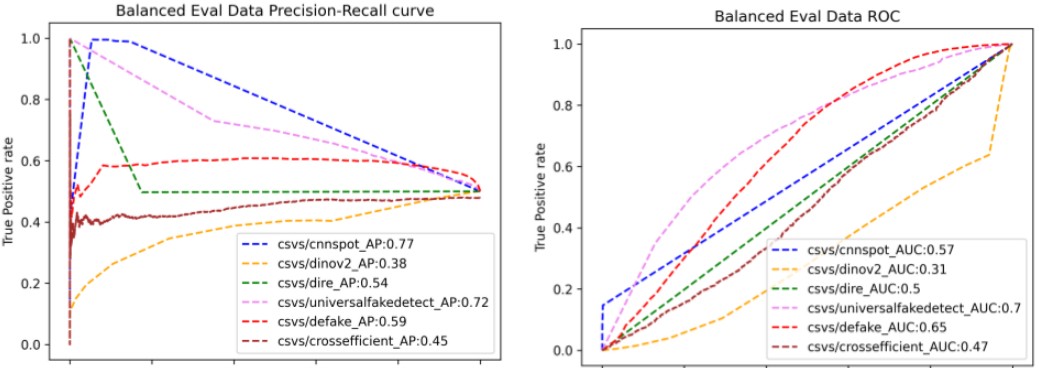

Figure 23: Precision-Recall curve with AP and AUC-ROC for each detector evaluated on a balanced Semi-Truths Eval Dataset

# G Compute Requirements

We used A40 GPUs from internal university cluster to run the augmentation techniques. Each dataset and diffusion model variation for each augmentation technique used 1 A40 GPU to run. Each image augmentation took ~2 minutes to generate for both prompt-based-edit and inpainting techniques.

# H License of Assets

1. LANCE [62]: Apache 2.0 license
2. Diffusers library (used for setting up inpainting pipeline and for all diffusion model inference) - Apache 2.0 license
3. LlaVa [50] - Apache 2.0 license
4. Llama [82] - Apache 2.0 license
5. UniversalFakeDetect [59] - No license
6. DIRE [88] - No license
7. DE-FAKE [73] - No license
8. DinoV2 [60] - Apache 2.0 license
9. CrossEfficientViT [11] - MIT License
10. CNNSpot [86] - Creative Commons Public License

