# OpenReview forum: "Semi-Truths: A Large-Scale Dataset of AI-Augmented Images for Evaluating Robustness of AI-Generated Image detectors"
_NeurIPS.cc/2024/Datasets_and_Benchmarks_Track — NeurIPS 2024 Track Datasets and Benchmarks Poster_

### Official Review · Reviewer_qbAq · 2024-07-12
**Paper review**

**Rating:** 6
**Confidence:** 4
**Clarity:** Some figures and tables lack proper l…

**Review:**

Please see the comments below.

**Strengths:**

- The paper addresses a timely issue in the field of AI, focusing on the robustness of image detectors against sophisticated AI-generated manipulations.
- The SEMI-TRUTHS dataset incorporates a large number of real and augmented images with comprehensive metadata.
- The methodology for generating augmented images and evaluating detector robustness is detailed and systematic.

**Additional Feedback:**

Please see the comments above.

**Correctness:**

The paper presents a coherent argument supported by data and methodological rigor. However, the lack of statistical validation of findings and insufficient correlation with human judgment are significant drawbacks.

**Documentation:**

No code or public database is available.

**Limitations:**

-  The data collection and augmentation methods might introduce biases that are not fully addressed in the analysis.
- Lack of detailed discussion on the limitations of the current state-of-the-art detectors and how SEMI-TRUTHS can specifically address these limitations.

**Opportunities For Improvement:**

- The methodology section needs a clearer explanation of the statistical methods used to validate the findings.
- Evaluation metrics and results should be better correlated with human judgment and real-world scenarios.
- The paper is heavy on technical jargon, making it less accessible to a broader audience. Simplification and clarification are needed.
- Literature review should more thoroughly address the limitations and potential biases of the SEMI-TRUTHS dataset.
- Ethical considerations of using such a dataset for potential misuse need deeper exploration.

**Relation To Prior Work:**

The paper provides a literature review but fails to adequately cover the limitations and potential biases of existing datasets and how SEMI-TRUTHS addresses these gaps. A more focused discussion on the advancements over prior work would strengthen the contribution.

**Summary And Contributions:**

The paper presents SEMI-TRUTHS, a dataset composed of 27,635 real images, 245,360 masks, and 850,226 AI-augmented images featuring varying degrees of edits. The research aims to evaluate the robustness of AI-generated image detectors against different augmentation methods, diffusion models, and data distributions. It highlights the varying sensitivities of state-of-the-art detectors to different types and degrees of edits, providing insights into their functionality.

---

> ### Author Rebuttal · Authors · 2024-08-17
>
> Thank you for your review ! We are glad you found our work presents a coherent argument supported by data and methodological rigor. Please allow us to address your concerns below.
>
> ### **Code availability:**
> We would like to clarify that the code is available in the supplementary material: semi_truths_process.tar.gz and the link to the dataset has also been provided in the supplementary: https://huggingface.co/datasets/semi-truths/Semi-Truths and https://huggingface.co/datasets/semi-truths/Semi-Truths-Evalset
>
> ### **Technical jargon:**
> Thank you for the recommendation. We have included a description of terms like scene complexity and diversity (Sec. 3, page 6), along with definitions for small, medium and large semantic changes (Table 2) and references for inpainting and prompt-based-editing (Sec. 2) among others. However, we understand if certain descriptions are still unclear or amiss and would greatly appreciate the reviewer’s guidance on terms needing further clarification. This will help us improve the clarity of our work and ensure a more pleasant reading experience for a broader audience.
>
> ### **Increasing literature review:**
> Thank you for your suggestion. We have discussed some limitations and potential biases of the Semi-Truths dataset in Sec. 5 along with discussing the limitations of existing datasets in Table 1. In addition, we would like to discuss that the dataset's biases stem from the inherent biases of the generative models used (LLMs and diffusion models). Since our pipeline utilizes these models off-the-shelf without fine-tuning, we do not introduce additional biases beyond those already present in the models. To address bias, we ensure diversity in edits by employing various diffusion models, LLMs, source benchmarks, and editing techniques, Furthermore, the metadata on edits (original and perturbed caption/label, original and edited images) provided by our dataset can help users investigate edit styles for unearthing commonly occurring biases in these generative models. We will include a discussion in the paper.
>
> ### **Ethical issues:**
> We would like to clarify that the ethical issues have been discussed in Sec.5. We would like to add that since the dataset was created from existing publicly available datasets containing real images with minimal potential of generating any harm, hence any manipulations on such images should not serve as a potential threat to society as per our knowledge.
>
> ### **Detailed discussion on SOTA detectors:**
> We would like to clarify that Sec.4 discusses the limitations of the current state-of-the-art detectors through an experimental approach by evaluating performance of these detectors on a general set of fake and real images (Table 3) followed by a detailed analysis of the model sensitivities. These experiments highlight the shortcomings of these detectors and how Semi-Truths was useful in unearthing these vulnerabilities which can help researchers improve these models by focussing on improving the training distribution (by using Semi-Truths or other favorable data points) or strategy to capture these vulnerabilities. We will include a detailed discussion addressing how Semi-Truths can benefit the community.
>
> ### **Statistical methods:**
> Thank you for your recommendation. We have included distributions for the different semantic change metrics, as well as the scene diversity and complexity distributions, with the corresponding binning cutoffs in the supplementary material (Sec D.2). These cutoffs were determined using a quantile binning approach on the entire dataset. Figure 5 in the supplementary provides qualitative examples alongside their quantitative quality check metric values, illustrating our quality check protocol. To provide further insight, we will include distributions for each quality check metric in the camera ready paper. Our quality check binning approach combines both qualitative and quantitative strategies, incorporating human judgment on the quantitative metrics. We will include further  discussion of this in the supplementary material and would greatly appreciate any guidance on whether certain statistical methods require further elaboration. If the reviewer has additional concerns pertaining to statistical validation that we have not addressed above, we would appreciate further clarification.
>
> ### **Evaluation metrics and their correlation with human judgment:**
> As shown in Table 5 and Figure 8, we find that our `Area Ratio` measure of image change has the greatest correlation with human perception. A number of the metrics we include in Semi-Truths that are widely used in Computer Vision, however, produce very low correlation coefficients. This does not inherently suggest that these measure are ineffective - as there may be an array of augmentations that would not stand out to a human observer. To further support this claim, we will include a few images in the appendix that demonstrates how an augmentation can be relatively large but hard for human annotators to notice (see example attached).

---

> > ### Comment · Reviewer_qbAq · 2024-08-29
> > **Thank you for the response**
> >
> > Thanks to the authors for the response. The reviewer has updated the score accordingly.

---

### Official Review · Reviewer_FB8v · 2024-07-23
**Novel and Comprehensive Dataset to Evaluate the Robustness of AI-Generated Image Detection**

**Rating:** 6
**Confidence:** 3
**Correctness:** The claims made in the paper seems co…
**Clarity:** Well written and mostly easy to follow.

**Review:**

This paper provides a valuable dataset compared to the existing ones. The contribution is valid and there is not clear flaw in the paper.

Please refer to the following sections for more details.

**Strengths:**

(1) The Semi-Truths dataset is both novel and comprehensive, offering a valuable benchmark for evaluating the robustness of AI-generated image detection models.

(2) The dataset's meta configuration is well-designed, with detailed descriptions of the dataset and the evaluation metrics provided by the authors to measure the strength of the image editing.

(3) The results demonstrate that existing AI-generated image detection models are not robust against the Semi-Truths dataset, highlighting its significance.

**Additional Feedback:**

Please refer to previous reviews.

**Documentation:**

The data description is clear and detail. The authors also provide the code for reproduction.

**Ethics:**

No clear ethical concern.

**Limitations:**

The paper has discussed its limitation.

**Opportunities For Improvement:**

(1) The term "AI-generated image detection" may not fully capture the paper's focus on image editing. The authors could clarify this definition or consider revising it to "AI-generated image editing."

(2) Many of the metrics in the paper are measured by AI models, which might not always be precise. For example, the strength of prompt editing is measured by LLAMA, which can sometimes make mistakes. To improve reliability, the authors could consider using a more comprehensive LLM, such as GPT-4 or Claude 3.5, or a combination of multiple models in future work.

(3) The human study section is not well explained, making it difficult to understand the differences between human evaluation and the proposed scoring function results. From Appendix E, it appears the human study does not align well with the automatic evaluation. The authors could provide more details on the human study to clarify this.

**Relation To Prior Work:**

This paper has discussed several state-of-the-art datasets and show its novelty.

**Summary And Contributions:**

This paper introduces a novel and comprehensive dataset called "Semi-Truths" to evaluate the robustness of AI-generated image detection. "Semi-Truths" refers to generated images that are modified versions of base images. The paper examines two image editing techniques using diffusion models: (1) Inpainting based on semantic masks, and (2) Prompt editing, which involves captioning the image and then editing it based on a revised caption. The authors include several metrics to measure the perturbation strength of inpainting and prompt editing, providing a thorough description of the dataset. The paper also includes a human study to align automatic evaluations with human perception. Additionally, it evaluates the robustness of existing AI-generated image detection models using the Semi-Truths dataset, showing that current models struggle with inpainting and prompt editing, underscoring the dataset's importance.

---

> ### Author Rebuttal · Authors · 2024-08-17
>
> Thank you for your review! We are glad that you find our dataset to be a valuable contribution to the community and value the novelty of our work.
>
> ### **Rephrase the title:**
> Thank you for the suggestion ! Yes, we agree and are proposing to change it to Semi-Truths: A Large-Scale Dataset of AI-Augmented images for Evaluating Robustness of AI-Generated Image detectors
>
>
> ### **LLM use, and metrics:**
> This is a great recommendation. We appreciate the opportunity to clarify that the majority of metrics are not dependent on LLM judgment. We've integrated a custom surface area-based change evaluation metric using traditional CV techniques to assess the ratio of image augmentation (Sec D.3). We also measure the degree of augmentation with MSE and SSIM. Additionally, scene diversity and complexity are evaluated through object count and diversity using human-annotated masks (Sec. D.2). For quality checks, we incorporate the BRISQUE score along with non-AI-based filtering techniques based on length and prompt attributes (Sec. 3.2 of the paper and Sec. C of the supplementary). We acknowledge that with our use of LlaMa and LlaVA in the pipeline, we also inherit its biases and potential pitfalls. However, one of the major advantages of LlaMA and LlaVA over other LLMs is that they are open source models. This ensures greater accessibility of the pipeline, while the modular structure also makes it easier to switch the LLM to any model or combination of models of the user's choosing. Additionally, we would like to add that we would incorporate a combination of more open source LLMs for computation of semantic change as part of future work.
>
> ### **Human Evaluation:**
> We agree that Sec. 4 does not communicate the motivations or conclusions of the user study clearly enough, and we thank the reviewer for pointing this out. We will revise the paragraph titled `Surveying Human Perception of Magnitudes of Change` in the following manner: "In this work, we leverage several algorithms to capture the degree of visual and semantic change achieved during image augmentation. To build an intuitive understanding of these metrics, we investigate if any of them correlate with how a person may perceive the magnitude of visual change. We conduct a user study where annotators classify the difference between pairs of original and augmented images into "not much", "some", and "a lot", corresponding to our "small", "medium", and "large" change bins. We then compute correlation coefficients (Pearson [38], Kendall Tau [61], and Spearman [1]) between human scores and quantitative measures in SEMI-TRUTHS. As shown in Table 5 and Figure 8, we find that our `Area Ratio` measure of image change has the greatest correlation with human perception. A number of the metrics we include in Semi-Truths that are widely used in Computer Vision, however, produce very low correlation coefficients. This does not inherently suggest that these measures are ineffective — as there may be an array of augmentations that would not stand out to a human observer." To further support this claim, we will include a few images in the appendix that demonstrates how an augmentation can be relatively large but hard for human annotators to notice (see example attached).

---

> > ### Comment · Reviewer_FB8v · 2024-08-26
> >
> > Thanks for the response during rebuttal! I don't have any additional question.
> >
> > I would like to retain my score and hope the authors could incorporate the suggested changes in their final version.

---

### Official Review · Reviewer_rPMV · 2024-07-24
**Review of Semi-Truths**

**Rating:** 7
**Confidence:** 4

**Review:**

The paper introduces a novel and useful dataset for evaluating the detection of AI-modified images that could be used for misinformation. The mostly automated pipeline for generating the dataset is well-described and includes multiple filters to ensure the quality and appropriateness of generated images. In addition, the variety of the dataset in terms of the various
- Data distributions
- Scene complexity/diversity
- Diffusion model
- Type of augmentation
- Magnitude of augmentation

allows the authors to perform an analysis comparing the performance of the various detection models with respect to each of these factors, potentially detecting witnesses/biases of these models.

While the experiments examining the different magnitudes of augmentation are an interesting independent contribution (e.g. finding that the Area Ratio correlates the best with human judgement), it is unclear how useful this is for the dataset/benchmark. Specifically, in Figure 7., it seems all models perform quite similarly for the different magnitudes of change.

Overall, the paper is strong and well-written but there are some issues with the dataset on HuggingFace (somewhat unclear instructions, missing inpainting folder) and the benchmark would benefit from a clearer protocol. I would be willing to increase my score if these are addressed.

**Strengths:**

- Comprehensive, automated pipeline with multiple filters to capture potential issues
	- Quality check, prompt filter, etc.
- Wide variety of image domains
- Multiple generative models used
- Analysis of factors potentially affecting performance of detection models

**Additional Feedback:**

N/A.

**Clarity:**

Overall, the paper is well-written and clear with useful figures for understanding the various pipelines used.

**Correctness:**

Overall, the approach described in the paper is sensible for automatically constructing the intended dataset and appropriate filters are used to ensure . At a glance, the code seems well-structured and relatively clean.

However, the eval set on HuggingFace is missing the inpainting folder which is concerning.

**Documentation:**

Dataset is available on HuggingFace. My initial instinct was to try to access it through the HuggingFace datasets API which didn't work. After, I tried downloading the tar files directly from HuggingFace which did work. Ideally, it should be made clearer how the dataset should be accessed from HuggingFace. Nonetheless, the structure of the dataset files is well described on HuggingFace and the various metadata fields are well described throughout the paper.

**Ethics:**

No significant ethical issues. Licensing of the dataset/other used datasets could be discussed further.

**Limitations:**

- The ratio of real to fake images is noticeably lower than other datasets (as a result, the evaluation set uses almost all the real images).
- Pipeline not completed automated yet (acknowledged by authors).

**Opportunities For Improvement:**

- Would be helpful to have a centralized table or piece of documentation describing the metadata fields instead of it being spread out throughout the paper.
- The benchmark part is unclear. If I were to create a new model to detect AI-edited images, it would be useful to have a standardized evaluation protocol and a more in-depth discussions of the metrics used for evaluating the models (e.g. of the precision/recall trade-off in this domain).
- Potentially include more examples of AI **modified** images being problematic (currently only includes the "Sleepy Joe" example which is a video and the Israel-Palestine article only seems to discuss images generated from scratch or being repurposed from other conflicts).

**Relation To Prior Work:**

Table 1 provides a succinct overview of other AI-generated image datasets. The novelty and advantages of Semi-Truths (focus on image-editing, the use of a variety of image domains and multiple generative models) are clearly described.

**Summary And Contributions:**

The paper proposes "Semi-Truths" as a new dataset/benchmark for evaluating the detection of AI-generated/modified images. Unlike most other datasets, they focus on the case of images being **edited** by some generative model (instead of images that are generated from scratch by some model). To do so, the dataset augments images to different degrees (and includes this information as metadata). Also, unlike the previous work, the datasets includes a wide variety of images (instead of just human faces) and uses multiple generative models for the editing.

---

> ### Author Rebuttal · Authors · 2024-08-17
>
> Thank you for your review! We are glad that you found the paper well written and clear and the approach to be sensible.
>
>
> ### **Clarity regarding the benchmark and metrics:**
> The benchmark created by the pipeline in Section 3.2 includes both real and AI-augmented images. The accompanying metadata file provides labels (real/fake) for each image, along with descriptive features and change metrics, such as native dataset, original and perturbed captions/labels, magnitude of change, etc. (see Metadata Structure [https://huggingface.co/datasets/semi-truths/Semi-Truths#metadata-structure] in the HuggingFace dataset card). Predictions from a new detector can be added as a separate column in this table, enabling performance evaluation across different change metric categories (magnitude of change, scene diversity/complexity, directional similarity).
>
> Since the real image class is unaffected in this analysis, the most informative metric to observe is Recall on the augmented class (we are adding support for additional metrics as discussed with P89j). A dip in Recall for a specific grouping would indicate the detector's sensitivity to that particular type of annotation. We will release a detailed step-by-step protocol and a notebook to facilitate the evaluation and analysis of any AI-augmented image detector. We will also include a clarification in the first paragraph of Section 4.
>
> ### **Data Imbalance:**
> The imbalance in our semi-truths dataset stems from pairing each real image with multiple perturbed variations using different magnitudes, techniques, and models. This design allows for a detailed exploration of model sensitivities across various dimensions, such as individual images, similar scenes, types of edits, their magnitudes, and augmentation techniques. Due to the dataset’s structure and objectives, this imbalance would persist even if more real images were added, as it would exponentially increase the corresponding fake images. To address this, we provide a dynamic pipeline that enables users to adjust the dataset according to their specific needs. For those requiring a more balanced dataset, we will include a script for easy integration of additional real images from source benchmarks on our Hugging Face dataset page. Additionally, we have recreated the experiment in Table 3 using a balanced dataset and reported the AUC-ROC (Fig. 1) and AUC-PR (Fig. 2). A discussion on these findings will be included in the paper.
>
>
> ### **Increase fake news examples:**
> Thank you for the suggestion. We do agree that increasing the number of real world examples of this issue will help emphasize how widespread this issue is and also the importance and relevance of the work. We will add these references related to AI-modified images spreading misinformation about the (1) entertainment industry, (2) religious community , (3) political candidates and cases where such fake stories resulted in (4) identity fraud and (5) explicit imagery.
>
> 1. https://www.nbcnews.com/tech/tech-news/katy-perrys-rihanna-met-gala-fake-ai-images-spread-rcna151163
> 2.  https://www.cbsnews.com/news/pope-francis-puffer-jacket-fake-photos-deepfake-power-peril-of-ai/
> 3. https://www.bbc.com/news/world-us-canada-68471253
> 4. https://www.forbes.com/councils/forbestechcouncil/2024/03/28/ai-is-the-final-blow-for-an-id-system-whose-time-has-passed/
> 5. https://www.latimes.com/california/story/2024-04-02/laguna-beach-high-school-investigating-creation-of-ai-generated-images-of-students
>
> ### **Dataset restructuring:**
> Thank you for the feedback on the dataset. We have addressed the concerns regarding the dataset in the following ways:
> - `Metadata`: We have attached a detailed metadata description in our HuggingFace dataset page in addition to Fig. 4 in the paper which highlights the metadata provided.
> - `Clarity`: We’ve updated our dataset card to include a lot more detail regarding the aim, background, and structure of the dataset. We renamed files and restructured the dataset to increase the clarity of the dataset [https://huggingface.co/datasets/semi-truths/Semi-Truths#semi-truths-dataset-a-large-scale-dataset-for-testing-robustness-of-ai-generated-image-detectors]
> - The dataset link https://huggingface.co/datasets/semi-truths/Semi-Truths contains individual folders for inpainting, prompt-based editing and original images and two individual metadata files for each image augmentation technique.
> - We have also updated the evaluation set (https://huggingface.co/datasets/semi-truths/Semi-Truths-Evalset) and the instructions on the datasets page for ease of understanding and use.

---

> > ### Comment · Reviewer_rPMV · 2024-08-19
> >
> > Thank you for your response. I appreciate the effort as the new dataset card/instructions/metadata descriptions are indeed helpful and improve the clarity of the benchmark. The additional examples also help motivate the benchmark. Given the explanation, the data imbalance makes sense (though the issue is perhaps not the imbalance but the relatively small amount of real images, which would be resolved with the new script). The additional metrics are valuable as well.
> >
> > I have increased my score from WA to Accept.

---

### Official Review · Reviewer_ybbx · 2024-07-27

**Rating:** 6
**Confidence:** 3
**Correctness:** yes
**Clarity:** yes

**Review:**

See Strengths and Opportunities For Improvement.

**Strengths:**

This paper is relatively well-motivated as AI-generated image detection is a crucial issue. I also find the evaluations thorough. The strengths are as follows.

The target issues of the paper are meaningful and worth exploring.
The motivation is clear.
The paper is easy to follow.

**Additional Feedback:**

None

**Documentation:**

yes

**Limitations:**

See Opportunities For Improvement

**Opportunities For Improvement:**

1. It is better to conduct an experiment using the images collected from fake news on the Internet.
2. The experiments with degraded images such as images with low resolution, and images with noise.

**Relation To Prior Work:**

yes

**Summary And Contributions:**

This paper introduces Semi-Truths, a dataset with a rich set of real and AI-augmented images, along with comprehensive metadata, designed for evaluating detector robustness across diverse editing techniques and data distributions. This paper indicates that current state-of-the-art detectors show sensitivity to the nuances of edits, suggesting the need for a deeper understanding of their functionality to improve detection accuracy.

---

> ### Author Rebuttal · Authors · 2024-08-17
>
> Thank you for your review! We’re glad you found the paper easy to follow and the topic meaningful. Recognizing the significance of the issue, we took great care to ensure the paper is as accessible as possible. We’re glad we are successful in doing so.
>
> ### **1. Experiments on real news stories:**
> We agree that it would indeed be an interesting experiment to use images from actual news stories. However, we are also concerned about the ethical issues raised by performing such an experiment. By creating a dataset of fake news generated by malicious actors, we believe we may be perpetuating harm to individuals or communities targeted by that content. The image augmentation pipeline we describe on pages 5 allows us to simulate this particular scenario at scale in a targeted and safe way.
>
> ### **2. Experiments with degraded images**
> Thank you for the suggestion. This is indeed an important issue that warrants attention. However, our research is specifically focused on understanding and investigating model sensitivities to semantic edits and magnitudes of change in high-fidelity images. A distinct line of research has produced a body of work that addresses this topic [1, 2, 3].
>
> **References:**
>
> [1] Haixu Song, Shiyu Huang, Yinpeng Dong, and Wei-Wei Tu. Robustness and generalizability of deepfake detection: A study with diffusion models, 2023.
>
> [2] Alexandros Haliassos, Konstantinos Vougioukas, Stavros Petridis, and Maja Pantic. Lips don’t lie: A generalisable and robust approach to face forgery detection. In Proceedings of the IEEE/CVF conference on computer vision and pattern recognition, pages 5039–5049, 2021.
>
> [3] Junyi Cao, Chao Ma, Taiping Yao, Shen Chen, Shouhong Ding, and Xiaokang Yang. End-to-end reconstruction-classification learning for face forgery detection. In Proceedings of the IEEE/CVF Conference on Computer Vision and Pattern Recognition, pages 4113–4122, 2022.

---

### Official Review · Reviewer_P89j · 2024-07-29
**The topic is interesting while the evaluation may need some improvements**

**Rating:** 6
**Confidence:** 3
**Correctness:** It is a dataset and the constructed p…
**Clarity:** Yes, the writing is good.

**Review:**

This paper attempts to address an urgent and meaningful question due to the increasingly popular of generative models. It is crucial to build robust models that could accurately identify real and fake/generated images.

The Pros:
1) The curated dataset includes a large number of fake images that have covered a variety of augmentations and have different manipulation levels---small/medium/large changes and have different scenes.

2) The curated dataset includes detailed meta-data and is in a large scale.

3) The authors have evaluated 6 SOTA detectors on their dataset.

4) The writing of this paper is good and it is easy to follow.

The Cons:
1) There is a significant imbalance ratio between the Real and Fake images.

2) Due to the nature of the imbalance in the dataset, the evaluation metrics used in experiments may not be super reliable.

**Strengths:**

In addition to the Pros I have listed in [Review],  I think the topic of this paper is quite interesting and is highly relevant to the community.

**Additional Feedback:**

N/A

**Documentation:**

Yes.

**Limitations:**

I have listed the limitations in the [Review].

**Opportunities For Improvement:**

I have listed the limitations in [Review]. I would suggest the authors: 1) curate a more balanced dataset (Real vs. Fake); 2) evaluate the detectors' performance with different metrics.

**Relation To Prior Work:**

Yes.

**Summary And Contributions:**

This paper attempts to answer the question: how robust and effectiveness of current AI-generated image detectors are to detect the AI generated images under various augmentations. To answer this question, the authors introduce a new dataset named SEMI-TRUTHS which contains 27,635 real images and 850,226 AI-augmented images. The authors find that the SOTA detectors are sensitive to different augmentations.

---

> ### Author Rebuttal · Authors · 2024-08-17
>
> Thank you for your review! We’re glad we were able to convey the relevance of the project and emphasize its importance. Please allow us to address the concerns raised in the review.
>
> ### **1. Data Imbalance**
> The observed imbalance arises because our semi-truths dataset pairs each real image with multiple perturbed variations, using different magnitudes, techniques, and models. This approach allows for granular investigation of model sensitivities at various levels, such as individual images, similar scenes, types of edits, their magnitudes, and the augmentation techniques. Given the dataset’s structure and objectives, this imbalance will persist, even with more real images, as it would exponentially increase the corresponding fake images. To address this, we offer a dynamic pipeline that allows users to adjust the dataset as needed for their specific problem. Additionally, for those users interested in a more balanced dataset for their particular use case, we will include a script for easy integration of additional real images from source benchmarks on our Hugging Face dataset page.
>
> ### **2. Evaluation metrics**
> We agree that the choice of metrics plays an essential role when evaluating on an imbalanced dataset. Therefore, we report individual class Precision, Recall and F1, along with overall Precision, Recall and F1, to provide a more interpretable overview. The majority of our experiments emphasize Recall on the fake class, as the impact of various augmentations primarily affects this class while leaving  real images largely unaffected. However, we recognize that additional metrics addressing class imbalance would offer a more comprehensive evaluation. Consequently, we have included AUC-ROC and AUC-PR curves for both the original (Fig 1.)  and a balanced(Fig 2.) evaluation set (containing 27,000 real and fake images) for the experiments mentioned in Table 3. We will also report these metrics for other experiments in the supplementary material.

---

> > ### Comment · Reviewer_P89j · 2024-08-29
> >
> > Thanks so much for answering my questions. I will retain my current rating for this version.

---

### Author Response · Authors · 2024-08-27

We sincerely appreciate the valuable suggestions and feedback provided by all the reviewers. We are pleased that our responses were able to address the concerns of reviewers **rPMV** and **FB8v**, and that they have contributed to a more favorable appraisal from reviewer **rPMV**. We look forward to hearing from the remaining reviewers and hope that our clarifications have successfully addressed the questions raised, further enhancing the overall evaluation. Thank you once again for your time and consideration.

---

### Decision · Program_Chairs · 2024-09-26

**Decision:**

Accept (Poster)

**Comment:**

The reviewers unanimously appreciated this paper. However, the contributions do not appear significant enough for an oral. In particular, the benchmark part of this paper is the main weakness.